# Tree mortality during long-term droughts is lower in structurally complex forest stands

Qin Ma[1,2,3], Yanjun Su [4,5,6] ✉, Chunyue Niu[4,5,6], Qin Ma[4,5,6], Tianyu Hu[4,5,6], Xiangzhong Luo [7], Xiaonan Tai [8], Tong Qiu [9], Yao Zhang [10], Roger C. Bales [11], Lingli Liu [4,5,6], Maggi Kelly[12,13] & Qinghua Guo [14,15]

Increasing drought frequency and severity in a warming climate threaten forest ecosystems with widespread tree deaths. Canopy structure is important in regulating tree mortality during drought, but how it functions remains controversial. Here, we show that the interplay between tree size and forest structure explains drought-induced tree mortality during the 2012-2016 California drought. Through an analysis of over one million trees, we find that tree mortality rate follows a "negative-positive-negative" piecewise relationship with tree height, and maintains a consistent negative relationship with neighborhood canopy structure (a measure of tree competition). Trees overshadowed by tall neighboring trees experienced lower mortality, likely due to reduced exposure to solar radiation load and lower water demand from evapotranspiration. Our findings demonstrate the significance of neighborhood canopy structure in influencing tree mortality and suggest that re-establishing heterogeneity in canopy structure could improve drought resiliency. Our study also indicates the potential of advances in remote-sensing technologies for silvicultural design, supporting the transition to multi-benefit forest management.

Global climate change is increasing the frequency and severity of droughts, and therefore threatening the health of forest ecosystems[1-3]. Taking the 2012–2016 drought in California, USA as an example, over 200 million trees were killed in the Sierra Nevada, a mountainous semi-arid region critical for California's water supply and goal of carbon neutrality[4,5]. These dead trees can significantly impact forest ecosystem processes, such as altering local hydrologic cycles through reducing the amount of evapotranspiration (ET) and increasing wildfire severity by providing more dead fuel loads[6,7]. Therefore, understanding drivers and mechanisms underlying tree mortality during drought has long been a research focus in forest ecology[8,9].

Tree death from drought is a multi-step ecological process[10]. Many factors can influence tree mortality during drought, and canopy structure has been identified as an important one[11,12]. Its influence on

[1]School of Geography, Nanjing Normal University, Nanjing 210023, China. [2]Key Laboratory of Virtual Geographic Environment (Nanjing Normal University), Ministry of Education, Nanjing 210023, China. [3]Jiangsu Center for Collaborative Innovation in Geographical Information Resource Development and Application, Nanjing 210023, China. [4]State Key Laboratory of Vegetation and Environmental Change, Institute of Botany, Chinese Academy of Sciences, 100093 Beijing, China. [5]China National Botanical Garden, 100093 Beijing, China. [6]University of Chinese Academy of Sciences, 100049 Beijing, China. [7]Department of Geography, National University of Singapore, Singapore 117570, Singapore. [8]Department of Biological Sciences, New Jersey Institute of Technology, Newark, NJ 07102, USA. [9]Department of Ecosystem Science and Management, Pennsylvania State University, University Park, PA 16802, USA. [10]Sino-French Institute for Earth System Science, College of Urban and Environmental Sciences, Peking University, 100871 Beijing, China. [11]Sierra Nevada Research Institute and School of Engineering, University of California, Merced, CA 95343, USA. [12]Department of Environmental Sciences, Policy and Management, University of California, Berkeley, CA 94720, USA. [13]Division of Agriculture and Natural Resources, University of California, Berkeley, CA 94720, USA. [14]Institute of Remote Sensing and Geographical Information Systems, School of Earth and Space Sciences, Peking University, Beijing 100871, China. [15]Institute of Ecology, College of Urban and Environmental Science, Peking University, 100871 Beijing, China. ✉e-mail: ysu@ibcas.ac.cn

drought-induced tree mortality can be analyzed from two aspects: individual tree size and neighborhood canopy structure (Fig. 1). Individual tree size (e.g., height, diameter at breast height, crown area) can influence tree mortality by regulating water demands and supply[13–16]. Despite extensive investigations into the influence of individual tree size on tree mortality during drought, particularly using tree height as an indicator, the results remain controversial. Many studies have found that taller trees are more hydraulically susceptible to drought[13,17], as they often possess longer water-transport pathways that could lead to accumulation of sapwood embolism and hydraulic failures[18]. Moreover, taller trees tend to exhibit disproportionately larger tree crowns than shorter ones due to allometric relationships of tree height with crown size and leaf area[19], which may further exacerbate their water demands during drought. However, allometric relationships of tree height with crown size and leaf area are species-specific, which may lead to distinct impacts on tree mortality during drought[20,21]. Additionally, larger trees may also have developed wider and deeper root systems, especially in harsh and fragile habitats with rocky terrains[22,23]. This enables them to access deeper water, increasing their survival rate during drought[14,24]. Therefore, a more comprehensive investigation of the relationship between individual tree size and tree mortality during drought among various tree species and environmental conditions is still urgently needed to understand the mechanisms underlying tree mortality.

As for the second aspect, neighborhood canopy structure can influence tree mortality during drought by regulating stand-level ecohydrological processes through resource competition and partitioning[15,25–27]. Trees in dense forests, particularly in monoculture settings, may experience intense symmetric competition for water during drought[5,6,12], leading to higher vulnerability to drought. For example, the mortality of ponderosa pine (*Pinus ponderosa*) trees

was found to have a positive exponential relationship with the stand-level tree basal area in western USA[28]. Moreover, increased asymmetric competition during drought in mixed-species forests may lead to larger trees acquiring water at the expense of smaller-stature trees, and therefore suppress the growth of smaller-stature trees[29,30]. However, dense forests with heterogeneous canopy structures can also create favorable microclimate and hydrologic conditions by altering within-canopy light environments[31,32]. Many studies found that tree mortality in denser forests, particularly the mortality of smaller-stature trees, is lower than that in open forests[33], and trees in denser forests have less growth reduction during drought[34]. As these mechanisms may vary with tree species and site conditions, the influence of neighborhood canopy structure on tree mortality during drought remains a subject of ongoing debate[14,35]. Examining trees over a large spatial scale may better inform how forests balance resource competition through neighborhood canopy structure, consequently influencing tree mortality during drought.

In this study, we explicitly examined the influence of canopy structure on tree mortality during drought. Two questions motivated our research. First, how does individual tree size (represented by tree height) relate to tree mortality across tree species during drought? Second, how does neighborhood canopy structure (represented by tree competition indices) influence tree mortality during drought? To address these questions, we used the 2012–2016 California drought as an example and delineated near 1.5 million individual trees from airborne light detection and ranging (lidar) data collected across a study area of approximately 150 km² in the southern Sierra Nevada. Tree species were recognized from pre-drought airborne lidar data and high-resolution aerial imagery collected in 2012 using a machine learning-based approach. Dead trees during the drought period were identified from airborne lidar data and multi-temporal

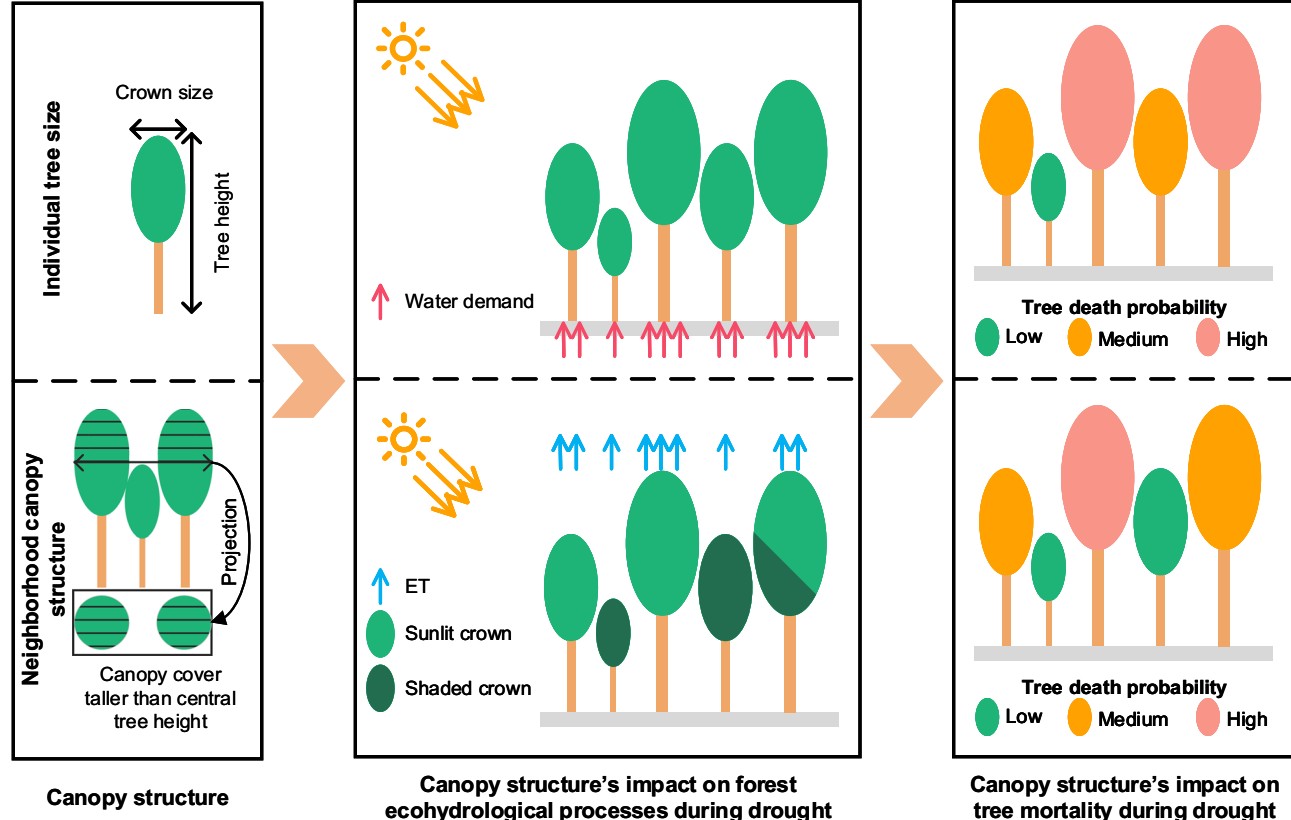

**Fig. 1 | A conceptual diagram of the influence of canopy structure on tree mortality during drought.** The top panel illustrates the potential influence of individual tree size on tree mortality during drought, and the bottom panel

illustrates the potential influence of neighborhood canopy structure on tree mortality during drought. ET represents evapotranspiration.

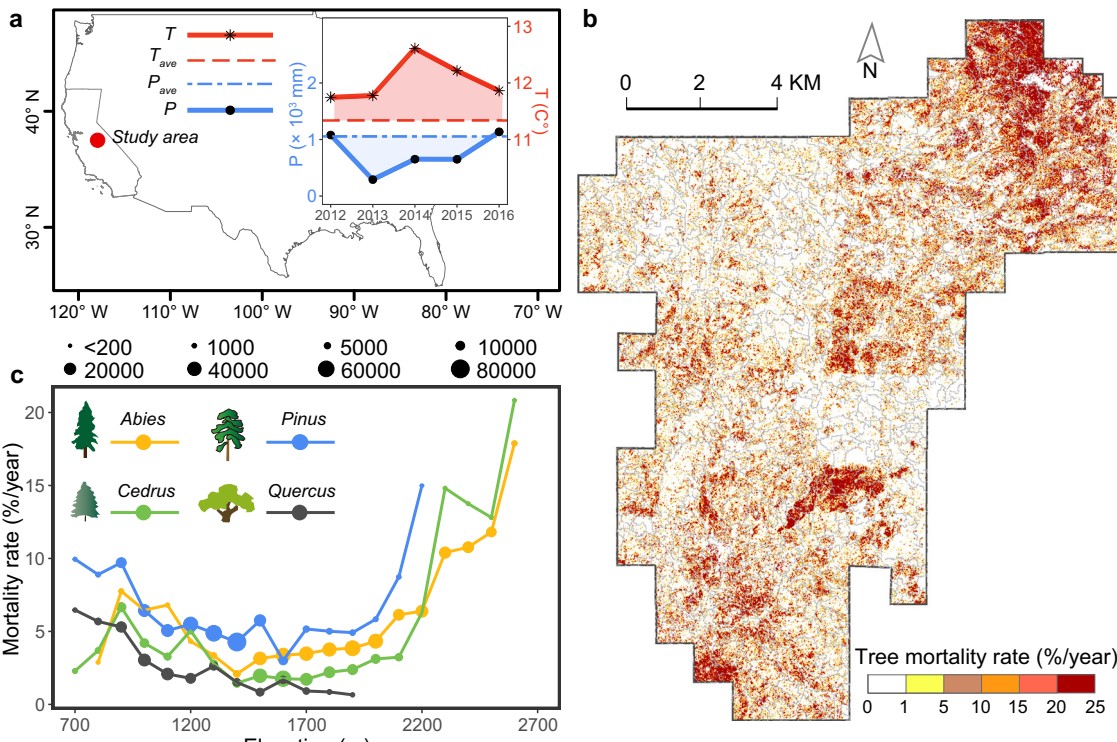

**Fig. 2 | Tree mortality rate varied by spatial location, genus, and elevation.**
**a** The location of our study area in the southern Sierra Nevada mountains, California, USA and its annual mean temperature (*T*) and annual total precipitation (*P*) during the 2012–2016 drought, compared to the average annual mean temperature (*T*ave) and the average annual total precipitation (*P*ave) from 1980 to 2019. Shaded areas indicate differences in precipitation and temperature between the drought period (2012-2016 in solid lines) and average conditions (in dashed lines). **b** Spatial pattern of tree mortality rate over 2012–2016 (100 m × 100 m cells) across the study area. A tree was identified as dead if over 35% of its crown was classified as dead, following the suggestion from Stovall et al.[17]. Polygons with gray boundary lines represent forest stands[61]. The mean stand size was 0.144 km² with a coefficient of variation of 86.6%. **c** Tree mortality rate by elevation for the four major genera in the study area. The marker size is proportional to the number of trees in each elevation gradient (100 m) for each genus.

high-resolution aerial imagery collected in 2012, 2014, and 2016, respectively. We found that tree mortality rate exhibited a "negative-positive-negative" piecewise relationship with tree height while consistently maintaining a negative relationship with neighborhood canopy structure.

## Results and discussion

### Impact of individual tree size on tree mortality

A total of 1,405,237 trees were segmented from the pre-drought airborne lidar data collected in the summer of 2012, and were classified into four genera (*Abies*, *Pinus*, *Cedrus*, and *Quercus*) with an overall accuracy of 66% and a kappa coefficient of 0.5 (Supplementary Fig. 1 and Supplementary Table 1). Each genus had distinguished habitats in the altitudinal spectrum, with *Abies* trees occupying the highest altitude, followed by *Cedrus*, *Pinus*, and *Quercus* trees (Supplementary Fig. 1c). During the 2012-2016 drought, the study area experienced a 294 mm drop in annual precipitation and a 0.70 °C rise in mean annual temperature compared to 1980–2019 averages (Fig. 2a). There were 180,765 trees out of the 1,050,960 trees taller than 5 m classified as dead during the 2012–2016 drought (F-score = 0.92), resulting in an average mortality rate of 4.3%/year (Supplementary Fig. 2 and Supplementary Table 2). *Pinus* trees exhibited the highest mortality rate at 5.3%/year (66,341 out of 312,930 trees), followed by *Abies* trees at 5.2%/year (78,069 out of 375,333 trees), *Quercus* trees at 2.7%/year (13,906 out of 128,764 trees), and *Cedrus* trees at 2.5%/year (23,393 out of 233,933 trees) (Supplementary Table 3). The overall mortality rate increased from 0.09%/year during the early drought (2012-2014) to 8.6%/year during the peak drought (2014-2016) (Supplementary Table 3). The tree

mortality rate for each genus generally decreased and then increased with elevation, except for *Quercus*, where the tree mortality rate decreased with elevation (Fig. 2c). It should be noted that dead trees were identified as those having over 35% of their crowns classified as dead[17]. Increase the threshold decreased the apparent mortality rate (Supplementary Table 3). To improve the robustness of the results, the mortality rates in our study area were calculated using thresholds of 30% to 50%, at a 5% increment, to validate the relationships between canopy structure and tree mortality rate.

To address the first question, we employed tree height as an indicator of individual tree size and examined its relationship with tree mortality rate. Mortality rate exhibited an overall "negative-positive-negative" piecewise relationship with tree height (Fig. 3a). That is, mortality rate decreased with height for trees under 14 m tall (coefficient of determination/$R^2$ = 0.98, *p*-value/*P* < 0.001), then increased with height for medium-sized (14–39 m) trees ($R^2$ = 0.96, *P* < 0.001), and decreased with height for trees over 39 m ($R^2$ = 0.67, *P* < 0.001) (Supplementary Table 4). The positive slope observed for medium-sized trees aligns with previous studies[17,36], suggesting that taller trees within this medium-height range may have a higher likelihood of experiencing hydraulic failure due to the accumulation of sapwood embolism during drought[10,19]. The negative relationship observed for smaller trees across all genera implies that they may be primarily influenced by competition. Small trees are often in their early life stage[36]. Young trees with relatively greater heights may possess more-established root systems than their smaller counterparts, especially in the rocky terrain of our study area. Consequently, they can effectively draw more water from the soil to mitigate the risk of hydraulic failure during drought[24]. Conversely, the negative relationship between tree

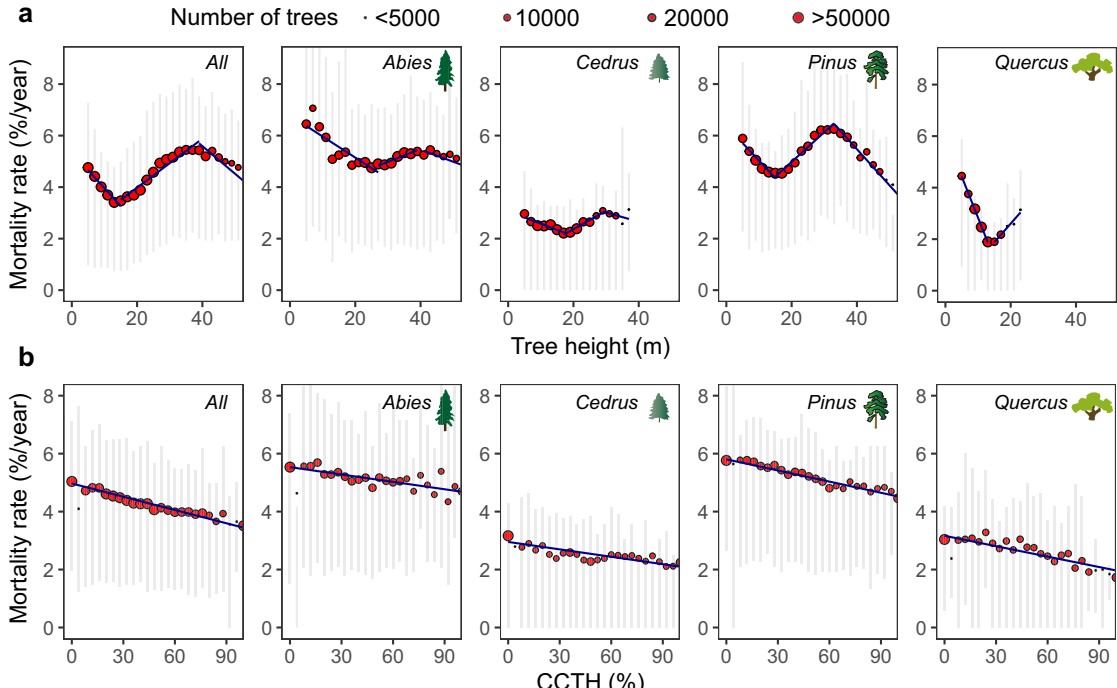

**Fig. 3 | Responses of tree mortality rate to tree height and canopy cover taller than central tree height (CCTH) during the 2012-2016 drought. a** Relationships between mortality and height for all genera (represented by *All*) and each respective genus. Tree mortality rate within each forest stand was binned by tree height with an interval of 2 m (number of bins = 54, 54, 34, 51, 20 for all genera, *Abies, Cedrus, Pinus,* and *Quercus,* respectively), and the stepwise linear regression method weighted by the number of trees in each stand was used to fit their piecewise linear relationships. **b** Relationships between tree mortality rate and CCTH of all genera (represented by *All*) and each respective genus. Tree mortality rate within each forest stand was binned by CCTH with an interval of 2% (number of bins = 50), and the linear regression method weighted by the number of trees in

each stand was used to fit their linear relationships. Only half of the bins were presented in the figure for visual clarity. Red dots in all subfigures represent the average tree mortality rate within each tree height or CCTH bin, and gray lines represent the range of tree mortality rate between the first and third quartiles of each bin. The size of red dots is proportional to the number of trees in each bin. Solid blue lines are fitted lines with a *p*-value (*P*) < 0.001. Results using different statistical methods (beta regression vs. linear regression), statistical units (forest stands vs. regular grids), percentage thresholds defining dead trees (from 30% to 50% with a 5% increment), and neighborhood sizes defining CCTH (15 m, 30 m, 50 m, and 100 m, respectively) are presented in Supplementary Figs. 4–7 and Supplementary Tables 4, 6–11.

height and mortality among larger trees likely arises because the largest trees tend to thrive in the most water-rich areas[37] and typically have developed wide and deep root systems with deeper zones of root-accessible water[14]. In our study area, we noted significant positive relationships between tree height and the topographic wetness index (TWI), and the correlation slope for tree height >34 m was over six times larger than that for trees <34 m tall (Supplementary Fig. 3 and Supplementary Table 5).

This piecewise height-mortality relationship could be observed in three out of the four genera (except for *Quercus*, which exhibited a "negative-positive" piecewise relationship). However, the tree height breakpoints between segments and their regression slopes varied by genus (Supplementary Table 4). These differences were consistent with site-level observations and might be attributed to species-specific physiologic traits and allometric relationships[20]. *Quercus* was the only genus that did not display the decreasing trend in the tallest tree group (Fig. 3a). This may reflect *Quercus* trees being water limited, growing mainly at lower elevations characterized by low precipitation and high temperatures, where water supply cannot support their growth to a very large heights[20,38] (Supplementary Fig. 1). Indeed, the highest *Quercus* tree in our study area was 24 m, lower than the breakpoints observed for other genera (29–40 m) (Supplementary Table 4). The decreasing trends for larger *Abies* and *Cedrus* trees were less noticeable than that of *Pinus* trees, likely because their soil and terrain conditions could not support the growth of large-sized trees (such as the *Abies* trees that dominant ridge tops and subalpine ecosystems) during drought[39] (Supplementary Fig. 1).

The observed piecewise tree height-mortality relationship remained consistent across variations in statistical methods (beta regression vs. linear regression; Supplementary Fig. 4a and Supplementary Table 4), statistical units (forest stands vs. regular grids; Supplementary Fig. 5a and Supplementary Table 6), and percentage thresholds defining dead trees (from 30% to 50% with a 5% increment; Supplementary Fig. 6a and Supplementary Table 7), demonstrating the robustness of our results. It is worth noting that the observed piecewise relationship has been reported in modeling-based studies[29,40,41]. Additionally, the negative tree height-mortality relationship for larger trees was also observed in Stovall et al.'s study[17], but was not reported because all trees taller than 30 m were grouped together in their "large" height class. Our analysis of live/dead conditions, with species information, and height measurements of over 1 million trees enables a more-detailed segmentation of the tree height-mortality relationship, underscoring the importance of big data in understanding the mechanisms contributing to tree mortality and improving the accuracy of tree mortality modeling during drought.

**Impact of neighborhood canopy structure on tree mortality**
Considering that the influence of neighborhood canopy structure on tree mortality is primarily mediated through resource competition and partitioning of space[15,25,26], we selected three widely recognized competition indices to address the second question. These indices included canopy cover taller than central tree height (CCTH), canopy cover taller than 66% of central tree height (CC66), and coefficient of variation in tree height (CVTH) (Supplementary Fig. 8)[42], which were

calculated using a neighborhood size of 15 m in radius. Given the greater significance of CCTH in explaining tree mortality rate compared to CC66 and CVTH (Supplementary Fig. 9), we employed CCTH as a representation of neighborhood canopy structure to explore the underlying mechanisms influencing tree mortality during drought. Overall, tree mortality rate demonstrated a significant negative relationship with CCTH ($R^2 = 0.86$, $P < 0.001$; Fig. 3b). A 10% increase in CCTH led to an average 0.17%/year decrease in tree mortality rate (Supplementary Table 8). Moreover, this negative relationship was observed across all genera ($R^2$ ranging from 0.31 to 0.76, $P < 0.001$), though with varying slopes (Fig. 3b). Specifically, a 10% increase in CCTH resulted in a 0.09%/year decrease in tree mortality rate for *Abies* trees, 0.07%/year for *Cedrus* trees, 0.18%/year for *Pinus* trees, and 0.10%/year for *Quercus* trees (Supplementary Table 8).

Growth among trees facing intense competition is often reported to be limited in normal climate conditions[42]. During drought, however, our results showed that higher CCTH benefits tree survival. We hypothesized that this effect might be because higher CCTH indicates more shading from nearby taller trees, resulting in lower leaf temperatures and reduced water loss through transpiration[43]. To test this hypothesis, we examined the relationships of CCTH and tree mortality with crown shadow ratio, defined as the average percentage of canopy shadow cast by neighboring trees during the daytime, simulated using a ray-tracing method. Figure 4a shows that CCTH exhibited a significant positive relationship with crown shadow ratio during the daytime (7:00 am to 6:30 pm) ($R^2 = 0.95$, $P < 0.001$). Each 10% increase in CCTH led to an average 1.1% increase in crown shadow ratio for a tree, thereby increasing the ratio of its shaded leaf area regardless of the crown shape (conifer or sphere shape). Using a process-based terrestrial biosphere model (i.e., Boreal Ecosystem Productivity Simulator), we quantified that a 10% increase in crown shadow ratio could lead to a 15% decrease in ET relative to trees without shadow in our study area (Supplementary Fig. 10). A strong negative relationship between crown shadow ratio and tree mortality during drought was observed in our study area ($R^2 = 0.77$, $P < 0.001$) (Fig. 4b). While crown shading would limit tree growth, the ET reduction brought about by crown shading could benefit survival of trees during drought. The hypothesis that CCTH influences tree mortality during drought by regulating crown shadow ratio was further validated through a structural equation modeling (SEM) analysis (Supplementary Fig. 11). Our findings emphasize that neither short nor tall trees are absolutely more vulnerable to droughts. Instead, the co-existence of short and tall trees might serve as a microclimate refugium for trees to survive drought stress.

Similar to the analyses regarding the impact of individual tree canopy structure on tree mortality during drought, we employed various statistical methods (beta regression vs. linear regression; Supplementary Fig. 4b and Supplementary Table 8), distinct statistical units (forest stands vs. regular grids; Supplementary Fig. 5b and Supplementary Table 9), varying percentage thresholds defining dead trees (from 30% to 50% with a 5% increment; Supplementary Fig. 6b and Supplementary Table 10), and different neighborhood sizes for calculating CCTH (15 m, 30 m, 50 m, and 100 m, respectively; Supplementary Fig. 7 and Supplementary Table 11) to conduct the aforementioned analyses. The relationships between CCTH and tree mortality rate were consistently negative across all genera. Further, the mixed linear modeling analyses incorporating both categorical and continuous tree height as random effects did not yield alterations in the observed relationships between CCTH and tree mortality (Supplementary Table 12).

Nevertheless, it is worth noting that the distinct definitions of different competition indices may lead to disparities in their relationships with tree mortality. For example, while similar negative relationships were observed for both CCTH and CC66 in relation to tree mortality rate, the relationships for CVTH varied for certain genera (*Abies* and *Quercus*) (Supplementary Fig. 12). Instead of displaying a negative relationship with tree mortality rate, CVTH exhibited a nonlinear relationship for *Abies* and a positive relationship for *Quercus* (Supplementary Fig. 12). In comparison to CCTH and CC66, which primarily focus on the central tree, CVTH encompasses height variations of all trees within a forest stand (Supplementary Fig. 8). Forest stands with high CVTH may not necessarily exhibit high CCTH and CC66. The unique environmental habitats and demographic characteristics of *Abies* and *Quercus* trees may potentially increase the likelihood of observing these mismatches. Specifically, a large proportion of *Abies* trees were situated at seasonally cold-limited high

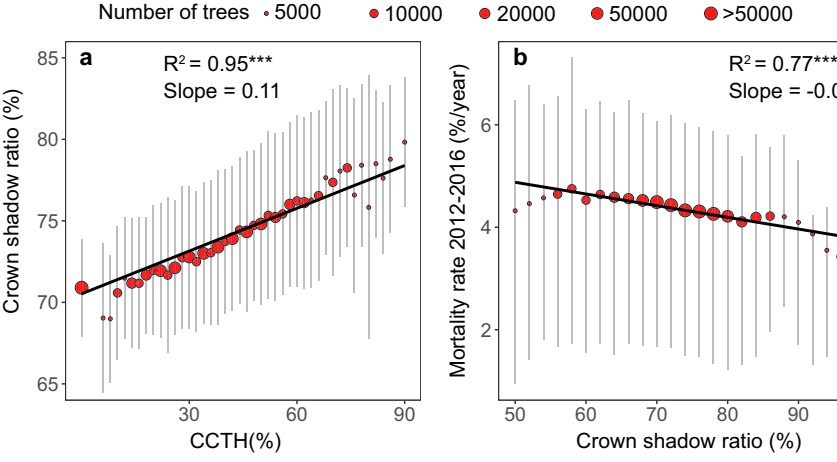

**Fig. 4 | Relationships among CCTH, crown shadow ratio, and tree mortality rate during the 2012-2016 drought. a** Relationship between CCTH and crown shadow ratio for all forest stands in the study area. Crown shadow ratio was defined as the average percentage of canopy shadow cast by neighboring trees during the daytime and was simulated through a ray tracing method. It was binned by CCTH with an interval of 4% in the weighted linear regression (number of bins = 43). **b** Relationship between crown shadow ratio and tree mortality rate during the 2012–2016 drought. Tree mortality rate for each forest stand was binned by crown shadow ratio with an interval of 2% (number of bins = 25). Red dots represent the average crown shadow ratio within each CCTH bin or the average tree mortality rate within each crown shadow ratio bin, while gray lines represent the range of crown shadow ratio or tree mortality rate between the first and third quartiles of each bin. The size of red dots is proportional to the number of trees in each bin. Solid black lines are significant fitted lines with a $P < 0.001$ (denoted as ***). The coefficient of determination ($R^2$) and slope of the fitted lines are reported. Results demonstrating the pathways of the influence of CCTH on tree mortality rate during drought by regulating crown shadow ratio through structural equation modeling are presented in Supplementary Fig. 11.

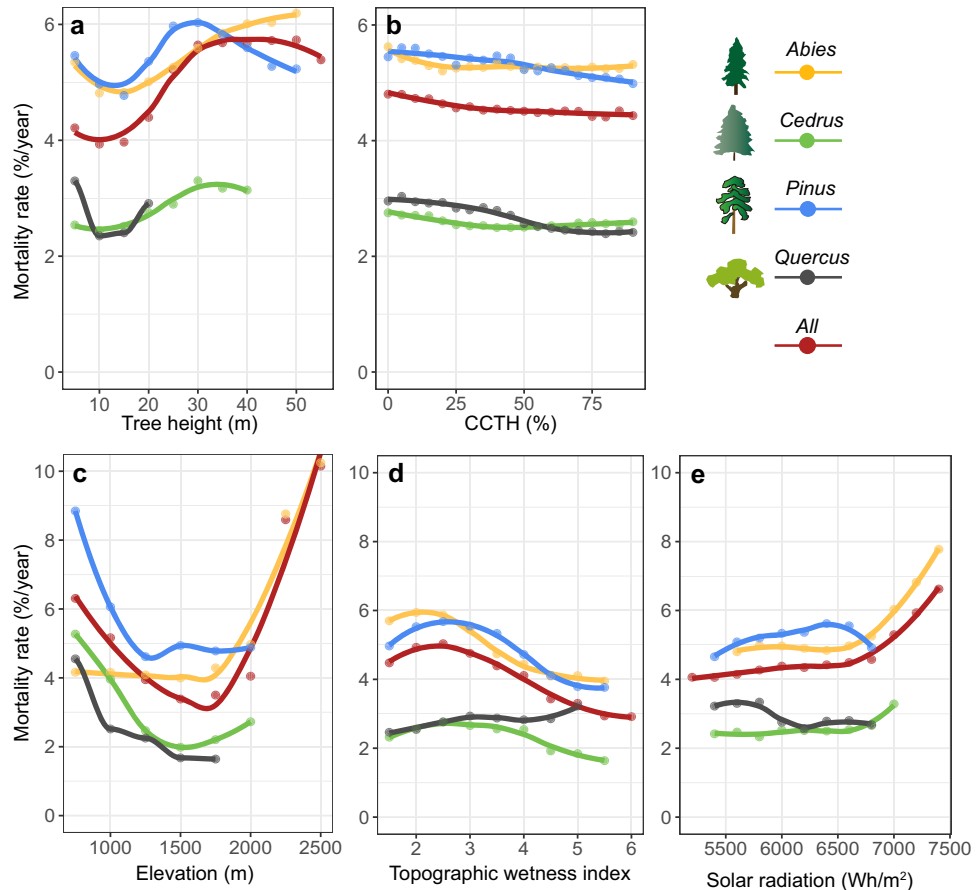

**Fig. 5 | Partial dependency values of tree mortality rate during the 2012-2016 drought to canopy structural attributes and environmental factors derived from random forest regression analyses. a** Responses of tree mortality rate to tree height of all genera (represented by *All*) and each respective genus. Tree mortality rate within each forest stand was binned by tree height with an interval of 5 m. **b** Responses of tree mortality rate to CCTH of all genera and each respective genus. Tree mortality rate within each forest stand was binned by CCTH with an interval of 5%. **c** Responses of tree mortality rate to elevation of all genera and each respective genus. Tree mortality rate within each forest stand was binned by elevation with an interval of 250 m. **d** Responses of tree mortality rate to topographic wetness index of all genera and each respective genus. Tree mortality rate within each forest stand was binned by topographic wetness index with an interval of 0.5. **e** Responses of tree mortality rate to solar radiation of all genera and each respective genus. Tree mortality rate within each forest stand was binned by solar radiation with an interval of 200 Wh/m².

altitudes (>2000 m), characterized by abundant bedrock outcrops with poor soil conditions, while the majority of *Quercus* trees grew in water-limited low altitudes with hot and dry climates (Supplementary Fig. 1). *Abies* and *Quercus* trees may adapt better to these harsh habitat environments[20,38,39], potentially making them taller than neighboring trees of other genera. As CVTH increases, the chance for *Abies* and *Quercus* trees to be taller than neighboring trees increases. The absence of shadows from taller neighboring trees may therefore lead to an increase in the mortality rate for *Abies* trees at high altitudes and for *Quercus* trees. A more comprehensive investigation into how different competition indices influence tree mortality rate, with the aid of detailed environmental and tree physiology observations, may further deepen our understanding of the mechanisms leading to tree death during drought.

**The synthesized effect of canopy structure and environmental factors on tree mortality**

To mitigate the coupled impact of canopy structure and environmental conditions on tree mortality rate during drought, we incorporated tree height and CCTH, alongside environmental factors (including elevation, TWI, and solar radiation), into random forest regression analyses to quantify their impacts on tree mortality rate. The random forest model, trained using both canopy structural attributes and environmental factors, accounted for 64% of the variation in

tree mortality rate within our study area during the 2012–2016 drought (Supplementary Table 13). The explanatory capacity of the model varied among genera, with the highest being for *Abies* ($R^2 = 0.67$), followed by *Pinus* ($R^2 = 0.44$), *Quercus* ($R^2 = 0.48$), and *Cedrus* ($R^2 = 0.42$) (Supplementary Table 13). When compared to the random forest models using only environmental factors (i.e., elevation, TWI, and solar radiation), the incorporation of tree height and CCTH accounted for a greater portion of the variation in tree mortality rate ($R^2$ increased by 6% to 13% for different genera), thus further highlighting the influence of canopy structure on tree mortality rate during drought (Supplementary Table 13).

Through the aforementioned random forest analyses, we derived the partial dependency values of tree mortality rate to each factor and found similar patterns between canopy structure and tree mortality rate, as reported above. The mortality rate for each tree genus decreased with height for small-sized trees, then increased with height for medium-sized trees, and finally decreased with height for large-sized trees (except *Quercus* and *Cedrus*) (Fig. 5a). Tree mortality rate consistently decreased with CCTH across all genera (Fig. 5b). In addition to the abovementioned canopy structural attributes, environmental factors were also related to tree mortality rate. Elevation showed a "U-shape" relationship with tree mortality rate (Fig. 5c), similar to the results shown in Fig. 2c; TWI had an overall negative relationship with tree morality rate (except *Quercus* trees) (Fig. 5d);

and solar radiation exhibited an overall positive relationship with mortality rate (except *Pinus* and *Quercus* trees) (Fig. 5e).

### Implications for forest management under a changing climate

Climate change is raising the frequency and intensity of chronic stresses and disturbances, not only causing widespread tree mortality in western USA, but also influencing forest health globally[44]. Forest restoration strategies primarily focusing on protecting big trees[45] may not increase the overall sustainability of forest ecosystems. The heterogeneity and diversity of tree sizes and age groups at the forest-stand level are vital for the sustainability of forest ecosystems, particularly in areas with frequent disturbance and drought stress. Tree species diversity often shows a positive relationship with canopy structural complexity[26,34], and the higher hydraulic diversity brought by high tree species diversity improves forest resilience to drought[46]. Our findings suggest that adopting a forest restoration strategy reestablishing heterogeneity in tree species diversity and canopy structural complexity can enhance forest resilience to extreme droughts[47].

While canopy structure has significant impacts on tree mortality, predicting tree mortality during drought must consider the compounding effects of canopy structure, environmental conditions, and interactions between biotic and abiotic factors[14]. This complexity may explain why most tree mortality prediction models have low explanatory power[14], and our study is no exception (Supplementary Table 13). Further improvements may require the combination of process-based modeling to simulate forests' mechanical responses to drought stresses and forest monitoring data over a wide range of spatial and temporal scales[9]. Advances in remote sensing technologies shed new light on forest monitoring, such as lidar for canopy structure monitoring, thermal remote sensing for canopy temperature monitoring, solar-induced chlorophyll fluorescence remote sensing for forest health monitoring, and microwave remote sensing for soil water content monitoring[48]. Although uncertainties may still exist in individual tree segmentation, species classification, dead tree detection, and forest attribute retrieval[49–51] (Supplementary Tables 1 and 2), integration with observations of tree physiological and structural traits as well as environmental factors may provide solutions to enhance our capability to monitor and predict tree mortality during drought[52]. While not yet deployed for designing silvicultural prescriptions, multi-objective modeling driven by new remote-sensing data is poised to offer added value for designing silvicultural prescriptions grounded in more direct measures of physiological and ecological processes[53]. Given the multi-billion-dollar challenge of restoring forests in the Sierra Nevada and the adoption of multi-benefit management over resource extraction[54], it is now timely to bring these tools from research to applications.

## Methods

### Study area

Our 151 km$^2$ study area is located in the southern Sierra Nevada mountainous forests (37°25′N, 119°36′W), ranging in elevation from 730 m to 2650 m (Fig. 2b, c). This area features a typical Mediterranean climate characterized by dry and warm summers as well as wet and cool winters. Between 2012 and 2016, the forests in this region experienced an extreme drought[4,5]. The average annual total precipitation over the past three decades was 984 mm, but decreased to 681 mm during 2012–2016 (Fig. 2a). The extreme drought was accompanied by significantly high temperatures (0.70 °C higher than the mean annual temperature for 1980–2019; Fig. 2a)[55], which accelerated snowmelt and prolonged the dry season[56]. Sierra mixed-conifer stands are the primary vegetation type in this region, with dominant tree species including *Calocedrus decurrens*, *Abies concolor*, *Pinus ponderosa*, *Pinus lambertiana*, *Quercus kelloggii*, and *Quercus virginiana*[57].

### Field measurements

Within the study area, a total of 121 field plots were set up following a stratified sampling approach aligned with established practices in forest inventory (Supplementary Fig. 13)[58]. Each plot was circular in shape with a radius of 12.62 m and an area of 500 m$^2$. The first plot was randomly chosen within the study area, and the remaining 120 plots were then placed on a 500-m spacing grid. If any plot was within 12.62 m of any landing, road, river, or otherwise physically inaccessible area, it was randomly relocated by 25 m along one of the four cardinal directions. These plots were surveyed in the summers of 2007 and 2008, and their locations measured using a Trimble GeoXH Global Positioning System (GPS) unit equipped with a Trimble Zephyr antenna (positioned at a height of 3 m). Continuously Operating Reference Stations and University NAVSTAR Consortium stations were available within 20 km of the designated plots for differential GPS post-processing. Throughout the data-collection phase, stringent efforts were devoted to ensuring a low positional dilution of precision (PDOP) value, rigorously maintained below 5. In instances where the PDOP value exceeded this threshold, immediate measures were taken to relocate the GPS receiver to a more open forest canopy location, typically up to 30 m away. To further enhance the GPS positioning accuracy, a minimum of 300 measurements were captured at one-second intervals for every position. Notably, the majority of positions encompassed over 1000 measurements. The culmination of these measures yielded centimeter-level precision accuracy in determining plot center locations. Once the precise coordinates of plot centers were established, we employed an Impulse laser ranger finder and an Impulse electronic compass to measure the distance and angle from the plot center to each individual tree, thereby georeferencing all trees with a diameter at breast height larger than 5 cm. In addition to the location of each tree, we also recorded its species information and measured its height using an Impulse laser ranger finder[58].

### Remote-sensing datasets

Two remote-sensing datasets covering the entire study area were collected, including airborne lidar data and high-resolution aerial imagery. The airborne lidar data were collected in the summer of 2012 using an Optech GEMINI airborne laser terrain mapper system, which was flown at a height of 600–1000 m above the ground. The lidar data had an average point density of 10 points/m$^2$ and were processed through a standardized procedure, including outlier removal, filtering, and normalization[49]. Outlier removal aims to mitigate the influence of noise points arising from wind, high-flying objects, and the multi-path effect. To achieve this, a distance-based method was employed, identifying noise points by assessing whether the average distance between a point and its ten closest neighboring points exceeds a threshold of $\mu + 5\sigma$ (where $\mu$ and $\sigma$ represent the mean and standard deviation of point distances, respectively)[59]. Filtering, the subsequent step, serves the purpose of separating ground points from non-ground points, facilitating the generation of terrain elevation products. In this study, we employed an enhanced progressive triangulated irregular network densification filtering algorithm[60]. Normalization, the final step, assumes the role of counteracting the influence of terrain elevation on lidar height measurements, accomplished by subtracting ground elevation from the elevation of a point. All preprocessing steps were executed within the LiDAR360 software (GreenValley International Inc.), employing the default parameter settings.

The high-resolution aerial images were collected by the National Agriculture Imagery Program (NAIP) in 2012, 2014, and 2016, respectively. They had a spatial resolution of 0.6–1 m and four spectral bands, including blue (435–495 nm), green (525–585 nm), red (619–651 nm), and near-infrared (808–882 nm). Here, we resampled all NAIP images to a 1-m spatial resolution and geo-registered them to airborne lidar data by manually collecting 69 road crossings as tie points (Supplementary Fig 13).

## Auxiliary datasets

Two types of auxiliary datasets were used in this study: forest stand boundaries and environmental factors. The boundary of 976 forest stands was obtained from Su et al. [61], who delineated areas with homogenous vegetation compositions and structures using the same airborne lidar data and NAIP images as in the current study. The mean forest stand size was 0.144 km$^2$ with a coefficient of variation of 86.6%. Five environmental factors−terrain elevation, slope, aspect, TWI, and solar radiation−were calculated from a digital terrain model (DTM) at 10-m resolution. The DTM was interpolated from lidar ground points using the ordinary kriging for interpolation. For this study, a typical summer day in the study area (August 1$^{st}$, 2016) was selected to represent the solar radiation condition, which was calculated as the sum of hourly solar radiations from 10 am to 2 pm. Hourly solar radiation was calculated from the lidar-derived DTM using the Area Solar Radiation tool in ArcGIS (ESRI)[62]. TWI, a function of slope and the upstream contributing area per unit width orthogonal to the flow direction[63], was calculated from the lidar-derived DTM using the TWI tool in SAGA-GIS.

## Individual tree segmentation and tree species classification

Dead trees were detected at the individual level through the combination of field data, lidar data, and high-resolution aerial imagery. To achieve this, we first delineated individual trees from lidar-derived canopy height model using a marker-controlled watershed segmentation algorithm[42]. Each detected tree was visually examined and manually corrected if necessary (see Supplementary Methods for details). A total of 1,405,237 trees were segmented within the study area. Despite the substantial manual work invested, we acknowledge that there still might be errors in the tree segmentation results, particularly for undercanopy small trees[49]. Given the potential segmentation errors of small trees and in identifying them from aerial imagery, we removed trees with a height <5 m in this study. Eventually, 1,050,960 trees were retained in subsequent analyses.

With the segmented individual trees, we implemented a random forest-based classification model to identify the species information of each tree from nine features, including one vegetation index derived from the pre-drought aerial imagery, four canopy structural features derived from the airborne lidar data, and four terrain features (see Supplementary Methods for details). A total of 977 field-surveyed trees were matched with the segmented trees through their geolocations (see Supplementary Methods for details), and half of them were randomly selected for training the classification model. Considering the difficulty in identifying and classifying tree species, we aggregated field-surveyed tree species into four genera, which are *Abies*, *Cedrus*, *Pinus*, and *Quercus*. All species-wise analyses were conducted at the genus level in this study. Using the trained classification model, we predicted the genus of each segmented tree, and the classification accuracy was evaluated by the remaining field-surveyed trees (Supplementary Table 1).

## Dead tree detection

To identify dead trees during drought from the segmented individual trees, we implemented a random forest-based classification model to predict six land-cover classes, which were dead tree, green tree, grass, bare ground, artificial object, and water (see Supplementary Methods for details). A total of 350 land patches with homogenous land cover, encompassing 61,965 NAIP pixels, were randomly selected and visually interpreted from the pre-drought (2012), early drought (2014) and peak-drought (2016) aerial imagery. Half of these pixels were used as the ground truth to train the classification model from five spectral features derived from the aerial imagery. The classification results were then overlaid with the segmented individual tree boundaries, and a tree was identified as dead if over 35% of its crown was classified as dead, following Stovall et al. [17]. If a tree was identified as dead in the

pre-drought aerial imagery, it was not considered a dead tree that occurred during the drought and was subsequently removed from the early drought and peak-drought dead tree detection results. To evaluate the accuracy of dead tree detection results, the remaining half of the visually interpreted pixels were used to determine the dead/live conditions of 635 trees using the same criterion as mentioned above. The recall, precision, and F-score were calculated for the dead tree detection results of each genus during each drought period. It is important to note that the dead tree detection results may vary based on the threshold used to determine dead trees. To ensure the robustness of the subsequent statistical analyses, we repeated the dead tree detection process by changing the threshold from 30% to 50% with a 5% increment. Each of the results was then used to evaluate the relationship between canopy structure and tree mortality rate.

## Canopy structural attribute extraction

To evaluate the influence of canopy structure on tree mortality during drought, two types of canopy structural attributes were extracted from the airborne lidar data, including individual tree size and neighborhood canopy structure. Individual tree size was represented by tree height, calculated as the maximum height above the ground within a tree segment. Since the influence of neighborhood canopy structure on tree mortality is primarily mediated through resource competition and area partitioning[15,25,26], it was represented by three widely recognized competition indices, including CCTH, CC66, and CVTH. These three indices can characterize the three-dimensional arrangement of canopy elements within the neighborhood of a targeted tree (a circular buffer with a 15-m radius in this study)[42]. CCTH and CC66 were calculated as the coverage of canopy elements within the neighborhood taller than 100% and 66% of a targeted tree, respectively, while CVTH was calculated as the coefficient of variation in tree height of all neighboring trees (Supplementary Fig. 8). To mitigate the influence of neighborhood size on the subsequent statistical analyses, we also calculated these neighborhood canopy structural attributes using different neighborhood sizes−15, 30, 50, and 100 m−and investigated their relationships with tree mortality rate during drought.

## Statistical analyses

To quantify tree mortality rate, we used both the abovementioned forest stands and regular grids (500 m × 500 m) as the basic statistical units, and the tree mortality rate of a statistical unit was calculated as the percentage increase of dead trees per year. To investigate the relationships between canopy structural attributes and tree mortality rate, either weighted linear regression analysis or weighted piecewise linear regression analysis was employed, and the slope, $R^2$, and $P$ values of each regression model were reported. The canopy structural attributes were binned for each 2-m difference in tree height, each 2% difference in CCTH, each 2% difference in CC66, and each 0.1 difference in CVTH within each statistical unit to simulate their responses to tree mortality rate; and the number of trees in each bin was used as the weight in the weighted regression analyses. As tree mortality rate is a value between 0 and 1, we also employed beta regression, a form of regression that takes dependent variable values ranging from 0 to 1[64,65], to validate the relationships between canopy structural attributes and tree mortality rate.

Considering the high relative importance of CCTH in explaining tree mortality rate, as determined by the percent increase in mean squared error derived from a random forest regression analysis[66], it was selected as an example to represent neighborhood canopy structural attributes in the statistical analysis hereafter (Supplementary Fig. 9). In analyzing variable importance, the tree mortality rate was calculated for each unit binned by the three stand-level canopy structural attributes (i.e., CCTH at a 5% interval, CC66 at a 5% interval, and CVTH at a 0.2 interval), and the random forest regression model was built by setting the number of trees (*ntree*) and the

number of features tried as each split (*mtry*) to 500 and 3, respectively. To avoid the coupled influence of individual tree size and neighborhood canopy structure on tree mortality rate, we further assessed the correlations between tree mortality rate and CCTH using mixed linear modeling, incorporating both categorical and continuous tree height as random effects. For the categorical random effect, tree height was divided into three groups, which were <33$^{rd}$ percentile, 33$^{rd}$ percentile - 66$^{th}$ percentile, and >66$^{th}$ percentile. To assess the significance of the random effect, we compared the mixed linear effect model for all genera or each genus with its corresponding model that did not account for tree height as the random effect, using analysis of variance. The slope and significance of the fixed effect (CCTH) and the significance of the random effect (tree height) were reported.

The combined influence of environmental factors and canopy structural attributes on tree mortality during drought was evaluated using the random forest regression. In addition to tree height and CCTH, we selected three environmental factors (i.e., elevation, TWI, and solar radiation) that showed observable relationships with tree mortality rate (Supplementary Fig. 15). While other environmental factors, such as temperature, precipitation, and vapor pressure deficit, were considered, they were not used in this study due to the lack of high-resolution products to represent individual tree-level variations[67], even though they may influence tree mortality during drought[17]. The tree mortality rate was calculated for each unit binned by the two canopy structural attributes (i.e., tree height at a 5-m interval and CCTH at a 5% interval) and the three environmental factors (i.e., elevation at a 250 m interval, TWI at a 0.5 interval, and solar radiation at a 200 Wh/m$^2$ interval). In this study, we built two random forest regression models for each genus and all genera combined, one using only environmental factors and one using both environmental and canopy structural factors. Each random forest regression model was built with *mtry* set to 3 and *ntree* set to 500. The relative importance of each factor was also quantified by the percent increase in mean squared error (Supplementary Fig. 15). The simulated response of tree mortality rate to each factor was presented using partial dependence values.

To further investigate the mechanism behind the influence of neighborhood canopy structure on tree mortality during drought, we simulated crown shadow ratio using the airborne lidar data. The crown shadow ratio of a tree was calculated as the average percentage of crown in the shadow cast by its neighboring trees within a 15-m radius neighborhood during the daytime. In this study, the shaded crown of each tree was simulated 23 times (simulated once every 30 min from 7 am to 6:30 pm) for a typical summer day of the study area (August 1$^{st}$, 2016) using a ray tracing method[68], and the average percentage of shaded crown during this period was used to represent crown shadow ratio (see Supplementary Methods for details). A similar weighted linear regression analysis, as mentioned above, was used to investigate the influence of neighborhood canopy structure (using CCTH as an example) on crown shadow ratio and the influence of crown shadow ratio on tree mortality during drought. Moreover, the SEM method, a multivariate modeling technique used to assess the casual connections among variables[69], was employed to validate the pathways linking CCTH to tree mortality rate through the regulation of crown shadow ratio. Furthermore, we simulated how the ET of a tree changed with crown shadow ratio using the Terrestrial Biosphere Model-BEPS model[70,71], and therefore explored the influence of changes in crown shadow ratio on tree-level water demand (see Supplementary Methods for details). We used both a conifer-shaped tree and a sphere-shaped tree as examples and simulated their relative changes in ET under different crown shadow ratios to that of a conifer-shaped or sphere-shaped tree without crown shadow during the daytime. Here, crown shadow ratio was set in the range from 73% to 92% with a step of 3%, corresponding to a ratio

between sunlit leaf area and the total amount of leaf area varying from 0.17 to 0.05 with a step of 0.02.

It should be noted that the *P* values for all coefficients obtained from linear regression were derived through a two-tailed *t*-test. All of the aforementioned statistical analyses were conducted using R through the packages *Betareg, randomForest, randomForestExplainer, pdp*, and *lavaan*.

### Reporting summary

Further information on research design is available in the Nature Portfolio Reporting Summary linked to this article.

## Data availability

The generated individual tree data with attributes of spatial locations, species, canopy structure, and live/dead conditions, as well as the processed data, are accessible on Figshare (https://doi.org/10.6084/m9.figshare.24278014).

## Code availability

The complete R code used for the calculation and visualization of the results is accessible on Figshare (https://doi.org/10.6084/m9.figshare.24278014).

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

## Acknowledgements

This study was supported by the National Key R&D Program of China grant 2022YFF0803100, the National Natural Science Foundation of China grant 42201366, the Nanjing Normal University grant 184080H202B34, the National Science Foundation grant 2106030, and the US Department of Energy grant DE-SC0023308. We acknowledge the data contribution of the Sierra Nevada Adaptive Management Project, an interagency project supported by the USDA Forest Service Region 5, USDA Forest Service Pacific Southwest Research Station, US Fish and Wildlife Service, California Department of Water Resources, California Department of Fish and Wildlife, California Department of Forestry and Fire Protection, and the Sierra Nevada Conservancy.

## Author contributions

Q.M.[1] and Y.S. designed the study; Q.M.[1], C.N., and Q.M.[2] performed the research; Q.M.[1] and Y.S. analyzed and visualized the data; Q.M.[1] and Y.S. wrote the original draft; and Y.S., Q.M.[1], C.N., Q.M.[2], T.H., X.L., X.T., T.Q., Y.Z., R.B., L.L., M.K., and Q.G. reviewed and edited the paper. Note that Q.M.[1] indicates the author Qin Ma from affiliations 1–3, and Q.M.[2] indicates the author Qin Ma from affiliation 4–6.

## Competing interests

The authors declare no competing interests.
