## [Peer Review File · Nature Communications]

Tree mortality during long-term droughts is lower in structurally complex forest standsREVIEWER COMMENTS

Reviewer #1 (Remarks to the Author):

SUMMARY

The paper titled, "Overshadowing effects in structurally complex forests reduce tree mortality during drought" evaluates the influence of tree and ecosystem structure on tree mortality in southern California during a period of extreme drought (2012-2016). The study specifically examines relationships between tree mortality and tree height versus the relative height of neighboring trees. The study highlights the importance of accounting for fine-scale structural heterogeneity in forest management and conservation. The main claims are as follows:

1. Tree mortality rates during a period of drought exhibited a nonlinear relationship with respect to tree height and neighborhood tree height for all four genera analyzed (Abies, Quercus, Cedrus, Pinus).
2. Trees with surrounded by taller trees (higher 'canopy cover taller than central tree height') experience higher shading which in turn results in decreased evapotranspiration, which could be beneficial for water conservation during droughts

This study offers an important contribution by demonstrating the role of spatial heterogeneity in forest structure in ecosystem responses to drought, building on previous work (e.g., Stovall et al 2019 Nature Communications). It leverages a unique dataset with over 1 million segmented trees and high resolution (60 cm) NAIP imagery. I do, however, have several concerns about the framing of the narrative and building on existing work, as well as some methodological questions, described in detail below.

Major Points

L176: The findings are not similar across all three metrics. Mortality patterns differ for trees in structurally complex forests (CVTH) versus trees overshadowed by neighboring trees (CCTH and CC66) versus just based on tree height

- There are actually interesting differences between the genera in terms of the relationship between mortality rates and CVTH. Cedrus and Pinus both show decreasing mortality rates with increasing structural heterogeneity (increasing CVTH), which is similar to the decreasing mortality observed with increasing overshadowing by neighboring trees (increasing CCTH and CC66). However, Abies and Quercus show increasing (Quercus) or nonlinear (Abies) trends in mortality rates with increasing structural heterogeneity. These results are worth unpacking further in the main manuscript. The nonlinear pattern for Abies is particularly interesting.
- I also suggest removing "in a structurally complex forest" in L37 from the abstract to avoid oversimplifying the story.
- The piecewise relationship between mortality and tree height is very interesting and warrants further discussion (L122-127). L129-134 start to do this, but why is there such a sharp change around trees 40 m tall? It's really clear pattern could be strengthened by more deeply linking the results to what we do and don't know physiologically and demographically about trees.
- L139: even the trees that were grouped into the 15-30m 'medium' category in Stovall et al 2019 exhibited increased mortality rates in 2016 though

Background literature

- The introduction would benefit from a more thorough overview of the ways that structure might influence tree mortality during droughts. There are definitely more than just the two ways described in L55-57. This includes going beyond tree size, which is very generic. I suggest reframing this section to think about tree level structure (e.g., tree level allometry with respect to height, diameter, crown architecture) and ecosystem scale structural heterogeneity, all of which could be further discussed. This would also allow for greater integration of the root allometry described in L61-63 and provide an explanation for the statement about heterogeneous forest canopies and more favorable microclimates and hydrological conditions in L73-75.
- Replace 'tree size' in L63 with 'tree and forest structure'
- The statement about studies on the relative importance of tree size in drought induced tree mortality being rare is not true. A quick search revealed:

- Bennett, A.C., McDowell, N.G., Allen, C.D. and Anderson-Teixeira, K.J., 2015. Larger trees suffer most during drought in forests worldwide. *Nature plants*, 1(10), pp.1-5.
- Mueller, R.C., Scudder, C.M., Porter, M.E., Talbot Trotter III, R., Gehring, C.A. and Whitham, T.G., 2005. Differential tree mortality in response to severe drought: evidence for long-term vegetation shifts. *Journal of Ecology*, 93(6), pp.1085-1093.
- Phillips, O.L., Van Der Heijden, G., Lewis, S.L., López-González, G., Aragão, L.E., Lloyd, J., Malhi, Y., Monteagudo, A., Almeida, S., Dávila, E.A. and Amaral, I., 2010. Drought-mortality relationships for tropical forests. *New Phytologist*, 187(3), pp.631-646.
- Wang, W., Peng, C., Kneeshaw, D.D., Larocque, G.R. and Luo, Z., 2012. Drought-induced tree mortality: ecological consequences, causes, and modeling. *Environmental Reviews*, 20(2), pp.109-121.
- Rather than claiming total novelty, this study would benefit from a more thoughtful explanation of how it's building on previous work to look at the relative differences in just tree height versus relative differences in tree height.
- In general, it would also be nice to see greater exploration of the results based on a deeper dive into the literature.
- L201: This isn't a new mechanism. It has been put forth in the literature before. For example:
 - Trugman, A.T., Anderegg, L.D., Anderegg, W.R., Das, A.J. and Stephenson, N.L., 2021. Why is tree drought mortality so hard to predict?. *Trends in Ecology & Evolution*, 36(6), pp.520-532.
 - Grote, R., Gessler, A., Hommel, R., Poschenrieder, W. and Priesack, E., 2016. Importance of tree height and social position for drought-related stress on tree growth and mortality. *Trees*, 30, pp.1467-1482.
- L253: This statement is not entirely accurate. If the authors stick with this argument, this sentence needs additional citations beyond ref. 32.

Clarify and report more methods and results in the main manuscript

- L452-464: The methods description for classifying a tree as dead warrants greater clarification. As written, it's unclear to me whether the tree being identified as dead if > 35% of its crown is dead constituted the training data for the model or not (L459-461). I was also unable to find where Stovall et al. 2019 (ref 14) states that they used the same method of identifying a tree as dead if over 35% of its crown was classified as dead. This seems like a very low threshold that could lead to large overestimation of tree mortality.
- L221: Report model performance

Minor Points

Abstract:

- L33 & L87: replace 1.5 million trees with 1 million trees since the analysis was actually done on 1,050,960 trees
- Remove 'drought-induced' from 'drought-induced tree mortality rates.' You cannot actually attribute causality. In addition, it's stated that the analysis was during the 2012-2016 drought in the previous sentence, which provides sufficient context.
- L39: Add 'neighborhood' to 'the importance of [neighborhood] forest structure'

Main:

- Replace 'size' with height throughout (e.g., L36, 65, 75, Figure 1 legend, etc.)
- I can understand why the authors use 'species groups' (e.g., L36, 96, 97, 223, 226) to refer to *Abies*, *Cedrus*, *Pinus*, and *Quercus*. However, 'genera' would be more appropriate.
- L63: Remove 'thus'
- L93: Report when the lidar data were collected in the main text
- L160: Also allometry and physiology in addition to species-specific traits
- L166: Add 's' at the end of 'Abie'
- L174: Add 'within a 15 m radius'
- L192: Define crown shadow ratio explicitly in main text
- L196: Define acronym first time it's used
- L265: Include reference for '...may explain why most tree mortality prediction models have low explanatory power'

Methods

- L411: Why 12.56 m diameter field plots?
- L421: Why not also include 2020 data given the lag in mortality post drought?

- L430-431: I'm guessing all of the topographic metrics were calculated from the airborne lidar data. State explicitly if so, otherwise report the data source
- L431: What is the data source for the solar radiation information?
- L437: I find it hard to believe that every single tree of the 1,405,237 segmented tree crowns were visually inspected and manually corrected.
- L447: Add 'and classifying' after identifying
- L451: Report classification accuracy in the main text, not just Supplementary Material
- L453: Replace 'land cover types' with 'categories' or 'classes'
- L472: Why a 15 m radius neighborhood area? Provide rationale. Was any sort of sensitivity analysis conducted to see how robust results are to neighborhood size?
- L478: Why use the forest stands as units of analysis as opposed to some more standardized grid? If the authors stick with the forest stands, report the minimum, mean, maximum, and coefficient of variation stand size.
- L497-503: Report model performance/evaluation. Did you use test and training data? K-fold cross-validation?
- L506-507: Is neighborhood still 30 m diameter?
- L509: Why Aug 1, 2016 and how sensitive is the model output to DOY?
- L516: Replace 'its' with 'tree-level' water demand
- L517: How do the results change if you use a sphere (which would be more appropriate for Quercus)?

Reviewer #2 (Remarks to the Author):

Overshadowing effects in structurally complex forests reduce tree mortality during drought.

This manuscript investigates the influence of canopy structure on drought-induced tree mortality during the 2012-2016 California drought. Authors combined rich data sources, field ground measurements, ALS acquisitions and aerial imagery, and different technical approaches to solve a given problem. From the methodological point of view constitutes a nice piece of research that I think is worthwhile. However, I have to indicate some concerns mainly related to the modelling approach and inference of the results, which prevents me from recommending publication in its current state, asking for some major changes. Furthermore, given the current form of the manuscript, there is missing information to reproduce the paper, and some methodological choices are unjustified. In the next paragraphs, I'll expose these main concerns and some proposals for improving the text.

Major issues:

Modelling approach. - Authors quantify tree mortality, as the tree mortality rate of a forest stand calculated as the percentage increase of dead trees per year. That means that the mortality rate is a percentage/proportion with a value between 0-100/0-1, bounded data. Then authors used a weighted linear regression or weighted piecewise linear regression to analyze the influence of tree size, relative size compared to neighboring trees (competition), and canopy structure on drought-induced tree mortality. However, I don't consider that a linear model is an appropriate approach for bounded data, as they may yield fitted values for the response variable that exceed its lower and upper bounds, below 0 and above 1, for example. Different regression models for modeling bounded data have been proposed; for instance, beta regression models (Ferrari and Cribari-Neto, 2004, <https://doi.org/10.1080/0266476042000214501>) are widely used. Therefore, I suggest to authors to rethink the modelling approach.

Inference. - besides revising the authors' modeling approach, I'm concerned about the inference that authors made evaluating the impact of tree size and relative tree size to neighboring trees on tree mortality. As authors claim in lines 53-57, "Many factors can influence tree mortality during drought, and canopy structure has been identified as an important one", so I found it very surprising that in such a complex and multifactor dependent process, authors decided to evaluate their research question fitting independent model for each one. Furthermore, I struggled to understand why authors analyzed the tree size and competition metrics in univariate mortality

models. Contradictorily, the authors combined the individual tree size and competition metrics, together with environmental factors, in the random forest regression analysis.

I don't consider that the inference and causality claimed in the sections 'Impact of absolute individual tree size on tree mortality' and 'Impact of relative tree size compared to neighboring trees on tree mortality' is supported by a robust analysis instead, the modelling is flawed. First, the models need to control for the effect of tree size and then evaluate the effect of competition. Also, adding the fact that there is not a comprehensive comparison of which of the competition metrics works better to explain tree mortality. Moreover, the authors don't report the fit statistics of the modelling and only report R2 for some models, which in any case, is not a good indicator of the model's performance or model selection. That is a basic step to define the importance and effect of the tree size or competition metrics in the tree mortality rate. Similarly, the analysis of the crown shadow ratio says very little by itself if it is not integrated with the modelling with other factors that simultaneously affect tree mortality. There are interacting and confounding effects between the explanatory variables that could affect the interpretation of the results and that the authors are not considering at all.

Lack of info details. - Authors need to include more details about the field plots measurements as they did in box1-4 for other aspects of the manuscript. There are some pieces of information missing that are relevant to understand the processing and the combinations of the different data sources used herein.

Specific comments:

L60. 'However, contradictory observations have also been reported'. Disagree with the statement, I consider that author fails the understanding the reference Stephenson et al., 2020. Stephenson et al.'s paper highlights the response in the magnitude of mortality and tree height relationship is species-specific (taxonomical groups). Moreover, Stephenson et al., showed that variation in mortality was greater within height classes (among taxonomic groups) than among height classes (within taxonomic groups). Thus, rather than a general relationship, a species-specific response is expected.

L61-63. This statement might be true for some specific conditions/sites. In the current state, the text sounds like a generally observed pattern, instead, I consider that ability to develop profound root systems is well known to be a species-specific trait and also depends on the growing conditions, i.e., soil type.

L65-68. There is strong feedback between canopy structure and the allocation of resources through competition ('relative size compared to neighboring trees') and the resulting standing biomass in a tree and stand. I advise to authors revise some fundamentals regarding the relationship between stand growth and stand structure to be included in the manuscript, for instance, Forrester 2019, <https://doi.org/10.1016/j.foreco.2019.05.053> and Pretzsch et al., 2015 <https://doi.org/10.1016/j.ecolmodel.2015.06.044>.

L73-73. Only true in relative terms, tree growth in denser stands is lower for a given tree size than in less dense stands due to competition and resources partitioning differences.

L76. I don't think the relationship between relative tree size and drought-induced mortality is rarely analyzed, maybe considering tree height as a measure of size. Authors could refer to the most recent textbook about Forest Growth Modelling to verify this statement. Plenty of examples of tree mortality dependencies on tree size, for instance, Weiskittel et al., 2011 - Forest Growth and Yield Modeling or Burkhardt & Tome 2012 - Modeling forest trees and stands.

L90. Pre-drought is 2012? Since this is the first time you declare pre-drought you should specify the year.

L113. In Fig2a the units of P(m) don't match to the text (mm) for example line 405. Also, the relationship between T and P doesn't show the drought years claimed, the authors should find a better way to illustrate the diagram.

L122-123. I don't consider that a "consistent positive or consistent negative correlation" with tree size should be expected. Typically, tree mortality has a 'U' relationship with tree size, i.e., tree diameter, in empirical mortality models, using logistic regression. Authors could refer to some studies such as Fortin, et.al., 2008 (doi:10.1051/forest:2007088) and Salas & Weiskittel, 2020 (https://doi.org/10.1016/j.foreco.2020.118369).

L131. Delete 'for water during drought'.

L176. I don't think that authors have similar findings between competition metrics here. For some species, the results seem quite different depending on the competition index, i.e., CCHH vs CVTH.

L406. What do you mean exactly by 'abnormally high temperature'? Please be specific and quantify.

L411. According to figure S6, looks like that a systematic sampling over a regular grid was used. It is not clear how the field plots were established.

L413. The authors need to include more details about the field plot measurements. Besides species ID and position, other measures were taken. Did the field crew identify live or dead trees for example?

L414 and 445. Not minor details are missing here. What is the precision of the tree location GPS data? For how long time the GPS placed to get the location of each tree? Did the tree location procedure include a GPS post-processed correction? Those are important details that should be included, to have more information about the precision in the co-registration data of field plots, lidar, and aerial imagery. I consider that the authors should extend the details as they did with the tree segmentation, species classification, and dead tree identification in boxes 1, 2, and 3.

L419-420. Can authors be more specific on the 'standardized procedure' used? Criteria to define and remove outliers, filtering and attributes for normalization.

L424. How many points did the authors use? Please add the location of the points in Figure S6.

L428-429. I suppose the authors used the same lidar and aerial imagery here, please clarify.

L431. How TWI and solar energy were derived from DEM?

L472. How do authors define the 15 m radius? Seems that is an arbitrary value that is properly justified. Looking at the distribution of tree height by species in Fig S1, there are some clear differences in the mean and distribution in tree height among species. The definition of the neighboring trees (competitive trees) certainly modifies the values of the CCTH, CC66, and CVTH used to evaluate the competition. Please, include an informative justification of the radius of provide evidence about the sensitivity of the competition metrics to the radius definition.

505. evapotranspiration (ET)

Box 1. 'Based on field measurements, segmented trees with a too small or a too large (>200 m²) crown area were probably caused by noise or mis-segmentation'. How much is too small, it is weird that there is a reference value for too large, but is missing for too small.

Box 2. 'Field records of individual trees were matched with lidar-derived individual tree crowns based on their spatial locations, and tree height and positioning uncertainty were also considered as constraints following the procedure used in Ma et al. (1)'. Please summarise how you consider the position uncertainty, given the interval between field ground survey and lidar acquisition.

Response to Reviewers

Reviewer #1

General comments

1. The paper titled, “Overshadowing effects in structurally complex forests reduce tree mortality during drought” evaluates the influence of tree and ecosystem structure on tree mortality in southern California during a period of extreme drought (2012-2016). The study specifically examines relationships between tree mortality and tree height versus the relative height of neighboring trees. The study highlights the importance of accounting for fine-scale structural heterogeneity in forest management and conservation. The main claims are as follows: (1) Tree mortality rates during a period of drought exhibited a nonlinear relationship with respect to tree height and neighborhood tree height for all four genera analyzed (*Abies*, *Quercus*, *Cedrus*, *Pinus*). (2) Trees with surrounded by taller trees (higher ‘canopy cover taller than central tree height’) experience higher shading which in turn results in decreased evapotranspiration, which could be beneficial for water conservation during droughts.

This study offers an important contribution by demonstrating the role of spatial heterogeneity in forest structure in ecosystem responses to drought, building on previous work (e.g., Stovall et al 2019 Nature Communications). It leverages a unique dataset with over 1 million segmented trees and high resolution (60 cm) NAIP imagery. I do, however, have several concerns about the framing of the narrative and building on existing work, as well as some methodological questions, described in detail below.

Response: Thank you for your valuable and positive feedback. Your insightful comments have been instrumental in enhancing the quality of our work. We have carefully considered each of your comments and made substantial revisions to the manuscript. Below, we provide comprehensive point-by-point responses to address the specific issues raised during the review process.

2. L176: The findings are not similar across all three metrics. Mortality patterns differ for trees in structurally complex forests (CVTH) versus trees overshadowed by neighboring trees (CCTH and CC66) versus just based on tree height. There are actually interesting differences between the genera in terms of the relationship

between mortality rates and CVTH. *Cedrus* and *Pinus* both show decreasing mortality rates with increasing structural heterogeneity (increasing CVTH), which is similar to the decreasing mortality observed with increasing overshadowing by neighboring trees (increasing CCTH and CC66). However, *Abies* and *Quercus* show increasing (*Quercus*) or nonlinear (*Abies*) trends in mortality rates with increasing structural heterogeneity. These results are worth unpacking further in the main manuscript. The nonlinear pattern for *Abies* is particularly interesting.

Response: We agree that the relationships of CVTH (coefficient of variation in tree height) with tree mortality rate diverged from those of CC66 (canopy cover taller than 66% center tree height) and CCTH (canopy cover taller than center tree height), and concur that investigating these distinctions presents an intriguing avenue for further research. As the reviewer pointed out, our analysis indeed revealed a nonlinear relationship between CVTH and tree mortality for *Abies* trees, while a slight positive association was evident for *Quercus* trees (Supplementary Fig. 12). These distinctions might be attributed to the disparities in their respective definitions. In comparison to CCTH and CC66, which primarily focus on the central tree, CVTH encompasses height variations of all trees within a forest stand (Supplementary Fig. 8). Forest stands with high CVTH may not necessarily exhibit high CCTH and CC66. The unique demographic characteristics of *Abies* and *Quercus* trees may potentially increase the likelihood of observing these mismatches.

Specifically, a large proportion of *Abies* trees were situated at seasonally cold-limited high altitudes (> 2000 m), characterized by abundant bedrock outcrops with poor soil conditions, while the majority of *Quercus* trees grew in water-limited low altitudes with hot and dry climates (Supplementary Fig. 1). *Abies* and *Quercus* trees may adapt better to these harsh habitat environments (Dolanc et al. 2013; Stephenson and Das 2020), potentially making them taller than neighboring trees of other genera. As CVTH increases, the chance for *Abies* and *Quercus* trees to be taller than neighboring trees increases. The absence of shadows from taller neighboring trees may therefore lead to an increase in the mortality rate for *Abies* trees at high altitudes and for *Quercus* trees.

We have further clarified the disparities between the relationships of CVTH with tree mortality and those of CCTH and CC66 in Lines 257-264, and we have discussed the potential reasons leading to these disparities in Lines 264-275. Moreover, we agree with the reviewer that comparing the impacts of different competition indices on tree

mortality rate can provide further insights on the results. This point has also been emphasized in Lines 275-278.

3. I also suggest removing ‘in a structurally complex forest’ in L37 from the abstract to avoid oversimplifying the story.

Response: We agree and have deleted it as suggested.

4. The piecewise relationship between mortality and tree height is very interesting and warrants further discussion (L122-127). L129-134 start to do this, but why is there such a sharp change around trees 40 m tall? It’s really clear pattern could be strengthened by more deeply linking the results to what we do and don’t know physiologically and demographically about trees.

Response: We appreciate the reviewer’s recognition of the intriguing piecewise relationship between tree height and tree mortality rate, which indeed warrants further discussion. As observed by the reviewer, we had discussed in-depth on the potential reasons to the piecewise relationship, and we agree there is a need of delving deeper to understand why there is a critical tree height breakpoint of 39 m.

Physiologically, tall trees with large canopies are postulated to be more vulnerable to drought-induced hydraulic stress due to their longer water transport pathways and higher water demands (Bennett et al. 2015). Following this hypothesis, Stovall et al. (2019) observed an increase in tree mortality during drought with increasing tree height, which aligns with our findings for trees with heights ranging from 14 m to 39 m. However, beyond a tree height of 39 m, our results reveal a decrease in tree mortality with further increases in height, which we suspect are relevant to the favorable environmental conditions in which these tall trees thrive.

Water availability serves as a critical determinant influencing the maximum height that trees can attain (Tao et al. 2016), often leading tall trees to flourish in the most water-rich landscapes. Even during drought, tall trees in water abundant areas still have a higher likelihood of accumulating more water (e.g., from melted snow in our study site) (Goulden and Bales 2019), which can potentially mitigate their hydraulic stress. To support our hypothesis, we examined the relationships between tree height the topographic wetness index (TWI) in our study site, and observed significant positive correlations between them. Interestingly, we found that the correlation slope after reaching a tree height of 34 m is more than six times larger than the slope observed before 34 m (Fig. R1), confirming the concentration of extremely tall trees

in areas with high water availability. The intriguing similarity in the tree height breakpoint of the tree height-TWI relationship (34 m) and that of the tree height-mortality relationship (39 m) highlights the potential critical role of the aforementioned mechanism in shaping the tree height-mortality relationship during drought.

It is worth noting that the tree height breakpoints of the piecewise height-mortality relationships vary among different genera (Fig. 3a and Supplementary Table 4). Nevertheless, these breakpoints generally align with the tree height breakpoints of the tree height-TWI relationships (40 m vs. 34 m for *Abies* trees, 33 m vs. 32 m for *Pinus* trees, and 29 m vs. 20 m for *Cedrus* trees). *Quercus* stands out as the only genus not displaying a decreasing trend in the tallest tree group (Fig. 3a). This distinction may be attributed to the fact that most *Quercus* trees grow in lower elevations, where water supply is limited due to low precipitation and high temperatures, potentially constraining their growth to larger sizes (Stephenson et al. 2019).

We have provided additional discussion on this topic in Lines 150-161 and Lines 172-188. Moreover, we have included a new figure and a new table in the supplementary materials (Supplementary Fig. 3 and Supplementary Table 5) that illustrates the correlations between tree height and TWI for different genera.

Fig. R1 Relationships between tree height and topographic wetness index (TWI)

for all genera and each respective genus. Forest stands were used as the basic statistical units here. TWI within each forest stand was binned by tree height with an interval of 2 m, and the piecewise linear regression method weighted by the number of trees in each bin was used to fit their relationships. Red dots in all panels represent the average TWI within each tree height bin, and grey lines represent the range of TWI between the first and third quartiles of each bin. Only half of the bins were

presented for visual clarity. The size of red dots is proportional to the number of trees in each bin. Solid blue lines are fitted lines with a p -value (P) < 0.1. The corresponding statistics of each regression segment are presented in Supplementary Table 5.

5. L139: even the trees that were grouped into the 15-30m ‘medium’ category in Stovall et al 2019 exhibited increased mortality rates in 2016 though.

Response: We agree with the reviewer’s observation that Stovall et al. (2019)’s study demonstrated an increase in tree mortality with tree height in the medium tree height group (from 15 m to 30 m). However, if we took a closer look at the Fig. 1a in Stovall et al. (2019), we noticed that the negative tree height-mortality relationship in larger trees was also observable. Unfortunately, this negative relationship did not stand out due to the grouping of all trees taller than 30 m into the “large” tree height category in their study. We have further clarified this point in Lines 182-188.

6. Background literature: The introduction would benefit from a more thorough overview of the ways that structure might influence tree mortality during droughts. There are definitely more than just the two ways described in L55-57. This includes going beyond tree size, which is very generic. I suggest reframing this section to think about tree level structure (e.g., tree level allometry with respect to height, diameter, crown architecture) and ecosystem scale structural heterogeneity, all of which could be further discussed. This would also allow for greater integration of the root allometry described in L61-63 and provide an explanation for the statement about heterogeneous forest canopies and more favorable microclimates and hydrological conditions in L73-75.

Response: We agree with the reviewer and followed the reviewer’s suggestion to rephrase the Introduction section to provide a more comprehensive overview on the potential influence of canopy structure on tree mortality during drought, starting from Line 55 to Line 91. We now refer to “absolute tree size” as “individual tree size”, and revised “relative tree size” into “neighborhood canopy structure”. Moreover, we have incorporated allometric relationships among tree height, crown architecture, and root traits, and their influences on tree mortality during drought in Lines 64-70.

7. Replace ‘tree size’ in L63 with ‘tree and forest structure’.

Response: In response to your previous comment, we have entirely rephrased the corresponding paragraph.

8. The statement about studies on the relative importance of tree size in drought induced tree mortality being rare is not true. A quick search revealed:

Bennett, A.C., McDowell, N.G., Allen, C.D. and Anderson-Teixeira, K.J., 2015.

Larger trees suffer most during drought in forests worldwide. *Nature plants*, 1(10), pp.1-5.

Mueller, R.C., Scudder, C.M., Porter, M.E., Talbot Trotter III, R., Gehring, C.A. and Whitham, T.G., 2005. Differential tree mortality in response to severe drought: evidence for long - term vegetation shifts. *Journal of Ecology*, 93(6), pp.1085-1093.

Phillips, O.L., Van Der Heijden, G., Lewis, S.L., López - González, G., Aragão, L.E., Lloyd, J., Malhi, Y., Monteagudo, A., Almeida, S., Dávila, E.A. and Amaral, I., 2010. Drought–mortality relationships for tropical forests. *New Phytologist*, 187(3), pp.631-646.

Wang, W., Peng, C., Kneeshaw, D.D., Larocque, G.R. and Luo, Z., 2012. Drought-induced tree mortality: ecological consequences, causes, and modeling.

Environmental Reviews, 20(2), pp.109-121.

Response: Thank you for providing the relevant references. We have thoroughly reviewed all the references you shared. These studies primarily focused on the susceptibility of large trees to drought, which falls under the category of individual tree size's influence on tree mortality, not about the influence of neighborhood canopy structure on tree mortality. Nevertheless, we agree with the reviewer that there are additional studies that have specifically investigated the influence of neighborhood canopy structure on tree mortality. We have rephrased the Introduction section, starting from Line 75 to Line 91, to provide an updated review of the impact of neighborhood canopy structure on tree mortality during drought. Moreover, we have also appropriately cited the references you provided in our manuscript.

9. Rather than claiming total novelty, this study would benefit from a more thoughtful explanation of how it's building on previous work to look at the relative differences in just tree height versus relative differences in tree height. In general, it would also be nice to see greater exploration of the results based on a deeper dive into the literature.

Response: We are sincerely grateful for your invaluable comments. Based on your feedback and in response to your previous comments, we have thoroughly re-written the introduction section to reflect the current knowledge of how canopy structure influences tree mortality during drought from the aspects of individual tree size and neighborhood canopy structure.

10. L201: This isn't a new mechanism. It has been put forth in the literature before.
For example:

Trugman, A.T., Anderegg, L.D., Anderegg, W.R., Das, A.J. and Stephenson, N.L., 2021. Why is tree drought mortality so hard to predict? *Trends in Ecology & Evolution*, 36(6), pp.520-532.

Grote, R., Gessler, A., Hommel, R., Poschenrieder, W. and Priesack, E., 2016. Importance of tree height and social position for drought-related stress on tree growth and mortality. *Trees*, 30, pp.1467-1482.

Response: Agree. We have rephrased the sentence to avoid confusion in Lines 242-245.

11. L253: This statement is not entirely accurate. If the authors stick with this argument, this sentence needs additional citations beyond ref. 32.

Response: We apologize for the potential confusion caused by the original statement. We have revised the sentence as follows:

“Forest restoration strategies primarily focusing on protecting big trees may not increase the overall sustainability of forest ecosystems.”

12. Clarify and report more methods and results in the main manuscript. L452-464: The methods description for classifying a tree as dead warrants greater clarification. As written, it's unclear to me whether the tree being identified as dead if > 35% of its crown is dead constituted the training data for the model or not (L459-461). I was also unable to find where Stovall et al. 2019 (ref 14) states that they used the same method of identifying a tree as dead if over 35% of its crown was classified as dead. This seems like a very low threshold that could lead to large overestimation of tree mortality.

Response: Following the suggestions of the reviewer, we have updated the relevant sections in methods and results. For constructing the training dataset, we also adopted a threshold of 35%, i.e., a tree was identified as dead if over 35% of its crown is classified as dead. We have further clarified this in Lines 480-483 and Supplementary Box 3.

In the study conducted by Stovall et al. (2019), they employed a similar threshold of 37.5% for tree mortality mapping in a southern Sierra Nevada forest, similar to our study site, because it provided the highest unbiased classification accuracy according to field samples (as illustrated in the Methods and Supplementary Figs. 1-4 in Stovall

et al. 2019). We adopted a similar threshold of 35% in our study but also acknowledge that using a relatively lower percentage threshold would result in an overestimation of tree mortality. To address this concern and assess the associated uncertainty in our results, we conducted a sensitivity analysis on the percentage threshold used to determine dead trees.

Specifically, we conducted an iterative analysis by changing the threshold from 30% to 50% with a 5% increment (i.e., 30%, 35%, 40%, 45%, and 50%). Each percentage threshold value was used to identify dead trees, and the relationships of tree mortality rate with tree height and CCTH were examined accordingly. As expected, we did find that the estimated mortality rate decreased with the choice of a higher threshold (Supplementary Table 3). Nevertheless, we found that the threshold selection had a minimal influence on our results, as the relationships between tree mortality rate and tree height, as well as CCTH, did not change significantly with the choice of thresholds (Fig. R4). Given these sensitivity analysis results and in alignment with Stovall et al.'s (2019) study, we continued to use 35% as the threshold value for determining dead trees in our study.

Moreover, based on the analysis above, we have added a new section to address the uncertainties incurred by the choice of the thresholds in Lines 126-128, Lines 179-180, Lines 249-250, Lines 481-483, and Supplementary Box 3. Additionally, we included Fig. R2 and its corresponding statistics in the supplementary materials (Supplementary Fig. 6 and Supplementary Tables 7 and 10) to further illustrate the sensitivity analysis conducted for different threshold values.

Fig. R2 Sensitivity analysis of the relationships between canopy structure and tree mortality rate during the 2012-2016 drought, considering the influence of the percentage threshold used to define dead trees. a, Relationships between tree height and tree mortality rate determined by a percentage threshold varying from 30% to 50% with an interval of 5%. Forest stands were used as the basic statistical units here. Tree mortality rate within each forest stand was binned by tree height with an interval of 2 m, and the piecewise linear regression method weighted by the number of trees in each bin was used to fit their relationships. **b,** Relationships between canopy cover taller than center tree height (CCTH) and tree mortality rate determined by a percentage threshold varying from 30% to 50% with an interval of 5%. Forest stands were used as the basic statistical units here. CCTH was calculated within a neighborhood size of 15 m in radius. Tree mortality rate within each forest stand was binned by CCTH with an interval of 4%, and the linear regression method weighted by the number of trees in each bin was used to fit their relationships. Red dots in all panels represent the average tree mortality within each tree height or CCTH bin, and grey lines represent the range of tree mortality rate between the first and third quartiles of each bin. Only half of the bins were presented for visual clarity. The size of red dots is proportional to the number of trees in each bin. Solid blue lines are fitted

lines with a $P < 0.05$. The corresponding statistics of each regression segment are presented in Supplementary Tables 7 and 10.

13. L221: Report model performance.

Response: The model performance has been added in Lines 302-308 of the manuscript.

Minor comments

1. L33 & L87: replace 1.5 million trees with 1 million trees since the analysis was actually done on 1,050,960 trees.

Response: Replaced as suggested.

2. Remove ‘drought-induced’ from ‘drought-induced tree mortality rates.’ You cannot actually attribute causality. In addition, it’s stated that the analysis was during the 2012-2016 drought in the previous sentence, which provides sufficient context.

Response: We agree and have deleted the phrase “drought-induced” throughout the manuscript.

3. L39: Add ‘neighborhood’ to ‘the importance of [neighborhood] forest structure’.

Response: Added.

4. Replace ‘size’ with height throughout (e.g., L36, 65, 75, Figure 1 legend, etc.).

Response: In response to your previous comments, we have now rephrased “relative tree size compared to neighboring trees” to “neighborhood canopy structure”.

5. I can understand why the authors use ‘species groups’ (e.g., L36, 96, 97, 223, 226) to refer to *Abies*, *Cedrus*, *Pinus*, and *Quercus*. However, ‘genera’ would be more appropriate.

Response: We agree and have replaced “species groups” with “genera” throughout the manuscript.

6. L63: Remove ‘thus’.

Response: The entire paragraph has been rephrased, in response to the general comment 6 of the reviewer.

7. L93: Report when the lidar data were collected in the main text.

Response: The airborne lidar data collection took place in the summer of 2012, as stated in the Methods section of the initial version of the manuscript. Following your suggestion, we have further emphasized this information in the Introduction section in

Line 107.

8. L160: Also allometry and physiology in addition to species-specific traits.

Response: We agree. We have revised the sentence in Lines 171-172.

9. L166: Add 's' at the end of 'Abie'.

Response: Corrected.

10. L174: Add 'within a 15 m radius'.

Response: Added as suggested.

11. L192: Define crown shadow ratio explicitly in main text.

Response: Crown shadow ratio was calculated as the average percentage of canopy in the shadow cast by its neighboring trees during daytime, as described in the Methods section of the initial manuscript. To enhance clarity, we have further emphasized this in Lines 234-237.

12. L196: Define acronym first time it's used.

Response: The definition of the acronym "ET" has been added in Line 51.

13. L265: Include reference for '...may explain why most tree mortality prediction models have low explanatory power'

Response: We have included a paper by Trugman et al. (2021) to support this statement in Line 350.

14. L411: Why 12.56 m diameter field plots?

Response: We apologize for the typographical error. The correct radius of the circular plot used in this study is 12.62 m, not 12.56 m in diameter. The field survey was conducted following the standard protocol of the Sierra Nevada Adaptive Management Project (SNAMP), a collaborative and innovative ecological research initiative aimed at addressing forest management challenges in the Sierra Nevada mountain range of California, USA. For this study, circular plots with a radius of 12.62 m were utilized to approximate a plot size of 500 m². Detailed information regarding the field plot design have been added in the Methods section in Lines 380-403.

15. L421: Why not also include 2020 data given the lag in mortality post drought?

Response: Thank you for the valuable suggestion. We concur that incorporating very-

high-resolution (VHR) images from 2020 would inform the continued tree mortality post drought. However, it is worth noting that around 35.71% of the study area was affected by a severe wildfire in 2017 (Ma et al. 2020). This complicates the differentiation between dead trees due to drought and those impacted by the wildfire, especially considering the potential combined influence of both wildfire and the legacy effect of drought (Stephens et al. 2018). Therefore, we chose not to include the VHR images of 2020 due to the difficulty in the attribution analysis.

16. L430-431: I'm guessing all of the topographic metrics were calculated from the airborne lidar data. State explicitly if so, otherwise report the data source.

Response: Yes, you are correct. All topographic metrics were calculated from the airborne lidar data. To avoid any confusion, we have provided additional clarification in Lines 435-438.

17. L431: What is the data source for the solar radiation information?

Response: The solar radiation information is calculated from the lidar-derived digital terrain information using the Area Solar Radiation tool in the ArcGIS (ESRI) software. We have further clarified this in Lines 433-436.

18. L437: I find it hard to believe that every single tree of the 1,405,237 segmented tree crowns were visually inspected and manually corrected.

Response: The visual examination and manual correction of 1,405,237 individual tree crowns is indeed a time-consuming and labor-intensive task. Starting in early 2017, we dedicated nearly three years to completing the processes of individual tree segmentation, tree species classification, and dead tree identification, with the assistance of three graduate students. The majority of our efforts were focused on ensuring the accuracy of the individual tree segmentation since it is the foundation for subsequent analyses. Moreover, we recognize the importance of developing a cross-region and cross-platform adaptive tree segmentation algorithm, which is capable of generating accurate results efficiently. We have further discussed this matter in the manuscript in Lines 445-451.

19. L447: Add 'and classifying' after identifying.

Response: Revised as suggested in Line 458.

20. L451: Report classification accuracy in the main text, not just Supplementary Material.

Response: We have reported the classification accuracy in Lines 110-112, as you suggested.

21. L453: Replace ‘land cover types’ with ‘categories’ or ‘classes’

Response: We have replaced “land cover types” with “land cover classes” in Line 465.

22. L472: Why a 15 m radius neighborhood area? Provide rationale. Was any sort of sensitivity analysis conducted to see how robust results are to neighborhood size?

Response: We understand the importance of justifying the selection of the neighborhood area. We selected a radius of 15 m based on our previous study (Ma et al. 2017), where we found that a neighborhood with a radius of 15 m effectively captures the competition among trees. In our study site, a medium-sized tree with a height of 20 m can cast a shadow ranging from 7.1 meters to 12.5 meters in length between 10 am and 2 pm on a typical summer day (August 1st, 2016). Even when considering the terrain effect, a 15-m radius is a suitable choice for effectively capturing the shadowing effect of neighborhood trees.

Nevertheless, we also recognize the significance of conducting a sensitivity analysis on the relationships between tree mortality and CCTH concerning the size of the neighborhood used for calculating CCTH. To address this, we calculated CCTH using four different neighborhood sizes with radii of 15 m, 30 m, 50 m, and 100 m, respectively, and evaluated their relationships with tree mortality. As expected, our results showed that variations in CCTH became smaller with the increase of neighborhood size (Fig. R3). Across all cases, the calculated CCTH of different neighborhood sizes showed consistently negative relationships with tree mortality, demonstrating the robustness of our results. We have added the content above to the manuscript in Lines 251-254 and Lines 496-499, and included Fig. R3 in the supplementary materials (Supplementary Fig. 7).

Fig. R3 Sensitivity analysis of the relationships between CCTH and tree mortality rate during the 2012-2016 drought, considering the influence of the neighborhood size used to calculate CCTH. CCTH was calculated using four different neighborhood sizes, with radii of 15 m, 30 m, 50 m, and 100 m, respectively. Forest stands were used as the basic statistical units here. Tree mortality rate within each forest stand was binned by CCTH with an interval of 10%, and the linear regression method weighted by the number of trees in each bin was used to fit their relationships. Red dots in all panels represent the average tree mortality within each tree height or CCTH bin. Only half of the bins were presented for visual clarity. The size of red dots is proportional to the number of trees in each bin. Solid lines are fitted lines with a $P < 0.05$. The corresponding statistics of each regression segments are presented in Supplementary Table 11.

23. L478: Why use the forest stands as units of analysis as opposed to some more standardized grid? If the authors stick with the forest stands, report the minimum, mean, maximum, and coefficient of variation stand size.

Response: We used forest stands as the basic statistical units of analysis because each forest stand exhibits relatively homogenous tree species composition, canopy structure, and environmental conditions. The forest stand map, created by the SNAMP science team using a combination of airborne lidar data, VHR aerial images, and environmental data (Su et al. 2016), has been employed to study the influence of forest treatment and environmental changes on forest hydraulic processes and wildlife habitats.

In response to the reviewer's comment, we also conducted an additional analysis using a regular grid with a size of 500 m \times 500 m as the basic statistical units. The relationships of tree mortality with tree height and CCTH remained consistent with

those obtained using forest stands (Fig. R4). Given its limited impact on our results, we continue to use forest stands as the basic statistical units in our main analysis, but also included the results obtained using the regular grid in the revised manuscript in Lines 178-179, Lines 248-249, and Lines 501-503, and added Fig. R4 in the supplementary materials (Supplementary Fig. R5). Additionally, we provided information on the minimum, mean, maximum, and coefficient of variation of stand size in Lines 138-139 and Lines 431-432.

Fig. R4 Relationships between canopy structure and tree mortality rate during the 2012-2016 drought using regular grids as the basic statistical units. a, Relationships between tree height and tree mortality rate of all genera (represented by panel *All*) and each respective genus. Tree mortality rate within each grid cell (500 m × 500 m) was binned by tree height with an interval of 2 m, and the piecewise linear regression method weighted by the number of trees in each bin was used to fit their relationships. **b,** Relationships between CCTH and tree mortality rate of all genera (represented by panel *All*) and each respective genus. Tree mortality rate within each grid cell (500 m × 500 m) was binned by CCTH with an interval of 4%, and the linear regression method weighted by the number of trees in each bin was used to fit their relationships. Red dots in all panels represent the average tree mortality within each tree height or CCTH bin, and grey lines represent the range of tree mortality rate between the first and third quartiles of each bin. Only half of the bins were presented

for visual clarity. The size of red dots is proportional to the number of trees in each bin. Solid blue lines are fitted lines with a $P < 0.05$. The corresponding statistics of each regression segment are presented in Supplementary Tables 6 and 9.

24. L497-503: Report model performance/evaluation. Did you use test and training data? K-fold cross-validation?

Response: As mentioned previously, we have already incorporated the performance of each random forest regression model (represented by R^2) in the revised manuscript in Lines 302-308 and Supplementary Table 13.

25. L506-507: Is neighborhood still 30 m diameter?

Response: Yes, you are correct. The neighborhood size remains 30 m in diameter to ensure consistency with the calculation of CCTH. We have made this clarification in Lines 548-549.

26. L509: Why Aug 1, 2016 and how sensitive is the model output to DOY?

Response: The reason we selected August 1st, 2016 as an example to simulate crown shadow ratio and evapotranspiration (ET) was because it is a representative summer day with the greatest radiation, highest temperature, and lowest water supply in our study area (Hopkinson and Battles 2015). As such, it serves as an ideal example for examining how spatial variations in canopy structure may impact the ecohydrological processes of forest ecosystems during drought.

While it is possible that the crown shadow ratio from the radiation transfer method and the simulated ET from the BEPS model are subject to seasonal variation, such as day of the year (DOY). However, the potential changes should be minimal compared to those caused by the diurnal cycle, because the variation in solar incidence angle (and thus crown shadow and ET) is primarily driven by the hour of the day rather than DOY in the dry summer. In our analysis, both the crown shadow ratio and ET were estimated at the hourly level—each half hour for crown shadow ratio and hourly for ET. The hourly results were then aggregated to daily to showcase how neighborhood canopy structure could change water demand and tree mortality during drought.

27. L516: Replace ‘its’ with ‘tree-level’ water demand.

Response: Replaced as suggested.

28. L517: How do the results change if you use a sphere (which would be more appropriate for Quercus)?

Response: Thank you for the valuable comment. We acknowledge that altering the shape of trees in the model may impact the simulation results. To address this concern, we conducted a new simulation analysis using a sphere-shaped tree specifically for *Quercus*. Similar to the simulation using the conifer-shaped tree, we calculated the relative changes in ET of a sphere-shaped tree under different crown shadow ratios compared to a sphere-shaped tree without any crown shadow during the daytime. The crown shadow ratio ranged from 73% to 92%, with a step of 3%, corresponding to a ratio between sunlit leaf area and the total leaf area ranging from 5% to 17%, with a step of 2%.

As depicted in Fig. R5, the simulation results using the sphere-shaped tree exhibited high similarity to the results obtained using the conifer-shaped tree. An increase in crown shadow ratio led to a decrease in actual ET relative to ET of trees without any shadow in our study area. We have provided detailed descriptions of these additional analyses in Lines 232-237, Line 558-561, Supplementary Box 4, and Supplementary Fig. 10.

Fig. R5 Responses of relative change in evapotranspiration (ET) to crown shadow ratio. The relative change in ET was calculated as the ratio between ET (ET_{shaded}) of a conifer-shaped or sphere-shaped tree under various crown shadow ratios (ranging from 73% to 92% with an interval of 3%) to that (ET_{reference}) of a conifer-shaped or sphere-shaped tree without crown shadow during the daytime (from 7:00 am to 6:30 pm) for a typical summer day (August 1st, 2016) in the study area. All

ET values were simulated using the Boreal Ecosystem Productivity Simulator (Supplementary Box 4). The red triangles represent the simulation results of a conifer-shaped tree, and the black dots represent the simulation results of a sphere-shaped tree. The solid lines represent the fitted lines with a P-value < 0.001 (denoted as ***), and the coefficient of determination (R^2) and slope values are reported.

References

- Bennett, A.C., McDowell, N.G., Allen, C.D., & Anderson-Teixeira, K.J. (2015). Larger trees suffer most during drought in forests worldwide. *Nature Plants*, 1, 15139
- Dolanc, C.R., Thorne, J.H., & Safford, H.D. (2013). Widespread shifts in the demographic structure of subalpine forests in the Sierra Nevada, California, 1934 to 2007. *Global Ecology and Biogeography*, 22, 264-276
- Goulden, M.L., & Bales, R.C. (2019). California forest die-off linked to multi-year deep soil drying in 2012–2015 drought. *Nature Geoscience*, 12, 632–637
- Hopkinson, P., & Battles, J. (2015). Learning how to apply adaptive management in Sierra Nevada forests: an integrated assessment. Final report of the Sierra Nevada Adaptive Management Project. *Center for Forestry, UC Berkeley*, <http://snamp.cnr.berkeley.edu/documents/676/index.html>. Accessed June
- Ma, Q., Bales, R.C., Rungee, J., Conklin, M.H., Collins, B.M., & Goulden, M.L. (2020). Wildfire controls on evapotranspiration in California's Sierra Nevada. *Journal of Hydrology*, 590, 125364
- Ma, Q., Su, Y., Tao, S., & Guo, Q. (2017). Quantifying individual tree growth and tree competition using bi-temporal airborne laser scanning data: a case study in the Sierra Nevada Mountains, California. *International Journal of Digital Earth*, 11, 485-503
- Stephens, S.L., Collins, B.M., Fettig, C.J., Finney, M.A., Hoffman, C.M., Knapp, E.E., North, M.P., Safford, H., & Wayman, R.B. (2018). Drought, Tree Mortality, and Wildfire in Forests Adapted to Frequent Fire. *Bioscience*, 68, 77-88
- Stephenson, N.L., Das, A.J., Ampersee, N.J., Bulaon, B.M., & Yee, J.L. (2019). Which trees die during drought? The key role of insect host-tree selection. *Journal of Ecology*, 107, 2383-2401
- Stephenson, N.L., & Das, A.J. (2020). Height-related changes in forest composition

- explain increasing tree mortality with height during an extreme drought. *Nature Communications*, *11*, 3402
- Stovall, A.E.L., Shugart, H., & Yang, X. (2019). Tree height explains mortality risk during an intense drought. *Nature Communications*, *10*, 4385
- Su, Y., Guo, Q., Fry, D.L., Collins, B.M., Kelly, M., Flanagan, J.P., & Battles, J.J. (2016). A Vegetation Mapping Strategy for Conifer Forests by Combining Airborne LiDAR Data and Aerial Imagery. *Canadian Journal of Remote Sensing*, *42*, 1-15
- Tao, S., Guo, Q., Li, C., Wang, Z., & Fang, J. (2016). Global patterns and determinants of forest canopy height. *Ecology*, *97*, 3265-3270
- Trugman, A.T., Anderegg, L.D.L., Anderegg, W.R.L., Das, A.J., & Stephenson, N.L. (2021). Why is Tree Drought Mortality so Hard to Predict? *Trends in Ecology & Evolution*, *36*, 520-532

Reviewer #2

General comments

1. This manuscript investigates the influence of canopy structure on drought-induced tree mortality during the 2012-2016 California drought. Authors combined rich data sources, field ground measurements, ALS acquisitions and aerial imagery, and different technical approaches to solve a given problem. From the methodological point of view constitutes a nice piece of research that I think is worthwhile. However, I have to indicate some concerns mainly related to the modelling approach and inference of the results, which prevents me from recommending publication in its current state, asking for some major changes. Furthermore, given the current form of the manuscript, there is missing information to reproduce the paper, and some methodological choices are unjustified. In the next paragraphs, I'll expose these main concerns and some proposals for improving the text.

Response: We sincerely appreciate your invaluable comments, especially your suggestions regarding the modeling approach and the inference of results. We have carefully considered all your comments and have made necessary revisions to improve our manuscript. Please see our point-by-point responses below.

2. Modelling approach. - Authors quantify tree mortality, as the tree mortality rate of a forest stand calculated as the percentage increase of dead trees per year. That means that the mortality rate is a percentage/proportion with a value between 0-100/0-1, bounded data. Then authors used a weighted linear regression or weighted piecewise linear regression to analyze the influence of tree size, relative size compared to neighboring trees (competition), and canopy structure on drought-induced tree mortality. However, I don't consider that a linear model is an appropriate approach for bounded data, as they may yield fitted values for the response variable that exceed its lower and upper bounds, below 0 and above 1, for example. Different regression models for modeling bounded data have been proposed; for instance, beta regression models (Ferrari and Cribari-Neto, 2004, <https://doi.org/10.1080/0266476042000214501>) are widely used. Therefore, I suggest to authors to rethink the modelling approach. Test the R beta regression models. Bound tree mortality for 0-1.

Response: We agree that tree mortality is a bounded value ranging from 0 to 1, making beta regression a more suitable method for predicting tree mortality based on

biotic and abiotic factors. Following your suggestion, we reanalyzed the relationships between tree mortality and canopy structure using beta regression. The beta regression results for the relationships between tree mortality and tree height, as well as CCTH, exhibited strong agreement with those obtained from linear regression across all tree genera (Fig. 3, Fig. R6, and Supplementary Tables 4 and 8). It is important to note that this study primarily aims to investigate the relationships between tree mortality and canopy structure rather than predicting tree mortality. We hope to highlight that in existing literature, linear regression is a widely adopted method for such analyses due to its ability to offer an intuitive understanding of the relationship between tree mortality and canopy structure through the use of the linear regression coefficient, although for predictive purposes, some other functions may serve better. Given the objective of our study is not prediction, we chose to continue using linear regression in the main results, but have included the beta regression analysis results in the supplementary materials and discussed this topic in Lines 176-178, Lines 246-248, and Lines 510-512 in the revised manuscript.

Fig. R6 Relationships between canopy structure and tree mortality rate during the 2012-2016 drought using beta regression. a, Relationships between tree height and tree mortality rate of all genera (represented by panel *All*) and each respective genus, using forest stands as the basic statistical units. Tree mortality rate within each forest stand was binned by tree height with an interval of 2 m. **b,** Relationships

between canopy cover taller than center tree height (CCTH) and tree mortality rate of all genera (represented by panel *All*) and each respective genus, using forest stands as the basic statistical units. Tree mortality rate within each forest stand was binned by CCTH with an interval of 4%. Red dots in all panels represent the average tree mortality within each tree height or CCTH bin, and grey lines represent the range of tree mortality rate between the first and third quartiles of each bin. Only half of the bins were presented for visual clarity. The size of red dots is proportional to the number of trees in each bin. Solid blue lines are fitted lines with a $P < 0.05$. The corresponding statistics of each regression segment are presented in Supplementary Tables 4 and 8.

3. Inference. – besides revising the authors' modeling approach, I'm concerned about the inference that authors made evaluating the impact of tree size and relative tree size to neighboring trees on tree mortality. As authors claim in lines 53-57, "Many factors can influence tree mortality during drought, and canopy structure has been identified as an important one", so I found it very surprising that in such a complex and multifactor dependent process, authors decided to evaluate their research question fitting independent model for each one. Furthermore, I struggled to understand why authors analyzed the tree size and competition metrics in univariate mortality models. Contradictorily, the authors combined the individual tree size and competition metrics, together with environmental factors, in the random forest regression analysis. I don't consider that the inference and causality claimed in the sections "Impact of absolute individual tree size on tree mortality" and "Impact of relative tree size compared to neighboring trees on tree mortality" is supported by a robust analysis instead, the modelling is flawed.

Response: We greatly value your insightful suggestion. It is indeed crucial to recognize the intricate nature of tree mortality during drought, which presents enormous challenge when attempting to predict tree mortality during drought through univariate analysis. Accurate prediction of tree mortality during drought must consider the compounding effects of canopy structure, environmental conditions, and interactions between biotic and abiotic factors. However, we wish to emphasize that the primary objective of our current study was to delve into the impact of canopy structure on tree mortality during drought and to uncover the underlying mechanisms. Consequently, our focus was directed towards elucidating relationships between

canopy structure and tree mortality rather than constructing predictive models for tree mortality.

Nonetheless, we also recognize the paramount importance of investigating these correlations while accounting for potential confounding factors. In response, we have taken comprehensive measures in the revised manuscript, including,

- 1) We incorporated random forest-based variable importance analysis to comprehensively compare the significance of various neighborhood canopy structural attributes on tree mortality.
- 2) We conducted mixed linear model analyses to assess the correlations between neighborhood canopy structural attributes while factoring in the influence of tree height.
- 3) We performed structural equation modeling analyses to validate the pathway through which neighborhood canopy structure modulates tree mortality by altering crown shadow ratio.
- 4) We introduced random forest-based partial dependence analyses to analyze the correlations between canopy structure and tree mortality, while also considering other relevant environmental factors.

Detailed descriptions and results of these analyses are elaborated in subsequent responses.

4. First, the models need to control for the effect of tree size and then evaluate the effect of competition.

Response: We agree that conducting additional analyses to explore the relationships between tree mortality and neighborhood canopy structural attributes, while accounting for the potential impact of individual tree size, would bolster the strength of our results. In response, we have introduced mixed linear modeling analyses to our revised manuscript. Specifically, we stratified trees into three tree height groups, i.e., low, medium, high, which was guided by equal numbers of trees (33.3% for each class in height gradient) within each genus. Then, we applied mixed linear modeling to examine the associations between tree mortality and neighborhood canopy structural attributes. In each model, neighborhood canopy structural traits (represented by CCTH) were treated as the fixed effect, while tree height was incorporated as the random effect. Our results demonstrated that the introduction of mixed linear modeling analyses did not yield alterations in the observed correlations between tree

mortality and CCTH (Table R1). In the revised manuscript, we provided the results of mixed linear modelling in Lines 254-256, Lines 520-526, and Supplementary Table 12.

Table R1 Statistics of relationships between CCTH and tree mortality rate during the 2012-2016 drought, considering the confounding influence of tree height. Tree height was divided into three groups, which were < 33rd percentile, 33rd percentile - 66th percentile, and > 66th percentile, and the tree height group information was used as the random effect in the mixed linear modeling analyses. Tree mortality rate within each forest stand was binned by CCTH an interval of 4%. The slope and P of the regression coefficient of CCTH were reported.

Genus	Slope	P
All	-0.014	<0.001
Abies	-0.010	<0.001
Cedrus	-0.007	<0.001
Pinus	-0.012	<0.001
Quercus	-0.014	<0.001

5. Also, adding the fact that there is not a comprehensive comparison of which of the competition metrics works better to explain tree mortality.

Response: We agree that a more comprehensive comparison of which neighborhood canopy structural attributes describing tree competition works better to explain tree mortality would further benefit the quality of the manuscript. In our original manuscript, we calculated three widely adopted competition metrics, i.e., CCTH, CC66, and CVTH, and analyzed their correlations with tree mortality individually. The results showed that CCTH had the strongest correlations with tree mortality among all genera and each respective genus (Supplementary Fig. 12 and Supplementary Table 8). In the revised manuscript, we added an extra variable importance analysis using random forest. The increase of mean-squared-error (%IncMSE) was used as the indicator for assessing variable importance. As shown in Fig. R7, CCTH was still the most important variable influencing tree mortality. Therefore, we continued to selecting CCTH as a representation of stand-level tree competition in our analyses. We have further discussed this in Lines 214-217 and Lines 513-520 and added the variable importance analysis results in the supplementary materials (Supplementary Fig.9).

Fig. R7 Relative importance of the three stand-level canopy structural attributes to tree mortality rate during the 2012-2016 drought. Variable importance was assessed by the increase of mean-squared-error (%IncMSE) derived from a random forest regression analysis.

6. Moreover, the authors don't report the fit statistics of the modelling and only report R^2 for some models, which in any case, is not a good indicator of the model's performance or model selection. That is a basic step to define the importance and effect of the tree size or competition metrics in the tree mortality rate.

Response: All regression analyses used in the current study (including piecewise weighted linear regression, weighted linear regression, beta regression, mixed linear model, and random forests) were provided with fit statistics, either in the figures or in the figure captions. Moreover, we have further systematically summarized all fit statistics in the supplementary materials (Supplementary Tables 4-13).

7. Similarly, the analysis of the crown shadow ratio says very little by itself if it is not integrated with the modelling with other factors that simultaneously affect tree mortality. There are interacting and confounding effects between the explanatory variables that could affect the interpretation of the results and that the authors are not considering at all.

Response: We agree that crown shadow ratio is an intermedia factor of canopy structure's influence on tree mortality. In fact, that was the primary purpose of introducing crown shadow ration into our study, which was to help to understand how CCTH influences tree mortality in a negative manner. Since it had a strong positive linear correlation with CCTH, we could not include it in the partial dependence analysis. In the revised manuscript, we added a structure equation modelling (SEM)

analysis to validate the paths that CCTH influences tree mortality through regulating crown shadow ration. As shown in Fig. R8, the path coefficients of the SEM model from CCTH to crown shadow ratio and from crown shadow ratio to tree mortality were significant ($P < 0.05$), and crown shadow ration consistently negatively contributed to tree mortality, similar to our linear regression results. In the revised manuscript, we have further reported these results in Lines 240-245, Lines 553-556, and Supplementary Fig. 11. We hope this additional analysis can delineate the link between canopy structure, crown shadow ratio and tree mortality.

Again, we did not aim to developing models to predict tree mortality in the current study. All statistical analysis used were aimed at explaining how canopy structure can influence tree mortality during drought. As stated by the reviewer, tree mortality during drought is a very complicated process. We believe this can also explain why our random forest models in Supplementary Table 13 had relatively low explanation power to tree mortality. We have discussed this in Lines 348-350.

Fig. R8 Pathways linking CCTH to tree mortality rate during drought through the regulation of crown shadow ratio derived from a structural equation modeling analysis. The structural equation modeling analysis was performed using R through the *lavaan* package. The coefficient (β) and P of each path are displayed, and the comparative fit index (CFI), Tucker-Lewis index (TLI), and standardized root mean square residual (SRMR) of the model are reported.

8. Lack of info details. - Authors need to include more details about the field plots measurements as they did in box1-4 for other aspects of the manuscript.

Response: Following the reviewer’s comment, we added more details about 1) field plot design including location and size in Lines 380-387; 2) the protocol for collecting plot and tree GPS locations in the field in Lines 387-402; 3) the steps to match field measured tree location and lidar tree segment in Supplementary Box 2. For reference, we also cited a paper from our colleagues who reported details about field survey design and implementation (Jakubowski et al. 2013).

Specific comments

1. L60. “However, contradictory observations have also been reported”. Disagree with the statement, I consider that author fails the understanding the reference Stephenson et al., 2020. Stephenson et al.’s paper highlights the response in the magnitude of mortality and tree height relationship is species-specific (taxonomical groups). Moreover, Stephenson et al., showed that variation in mortality was greater within height classes (among taxonomic groups) than among height classes (within taxonomic groups). Thus, rather than a general relationship, a species-specific response is expected.

Response: We apologize for the potential confusion caused by our statement. We totally agree that Stephenson et al.’s (2020) study underscored the significance of exploring species-specific correlations between tree height and tree mortality, and it is a crucial piece that motivated us to diligently ascertain the genus information for each segmented tree in our study. Our results generally align with the conclusions drawn by Stephenson et al. (2020). Although a prevailing “negative-positive-negative” pattern was characterized by the piecewise relationships between tree height and tree mortality (excluding *Quercus* trees), it is noteworthy that the breakpoints for these piecewise relationships varied across genera. In response to your comment and a similar comment from the first reviewer we have rephrased this paragraph in Lines 55-74, to avoid any potential confusion.

2. L61-63. This statement might be true for some specific conditions/sites. In the current state, the text sounds like a generally observed pattern, instead, I consider that ability to develop profound root systems is well known to be a species-specific trait and also depends on the growing conditions, i.e., soil type.

Response: We agree that the capacity to establish robust root systems is highly dependent on species-specific traits and growth conditions. As mentioned earlier, we have rephrased the entire paragraph in Lines 55-74 to avoid any potential ambiguities.

3. L65-68. There is strong feedback between canopy structure and the allocation of resources through competition (“relative size compared to neighboring trees”) and the resulting standing biomass in a tree and stand. I advise to authors revise some fundamentals regarding the relationship between stand growth and stand structure to be included in the manuscript, for instance, Forrester 2019,

<https://doi.org/10.1016/j.foreco.2019.05.053> and Pretzsch et.al., 2015

<https://doi.org/10.1016/j.ecolmodel.2015.06.044>.

Response: Thank you for your insightful comment and the recommended references. We have revised the entire paragraph in Lines 75-91 and included the suggested references.

4. L73-73. Only true in relative terms, tree growth in denser stands is lower for a given tree size than in less dense stands due to competition and resources partitioning differences.

Response: Agree. We have rephrased the entire paragraph. Please see our response to your comments above.

5. L76. I don't think the relationship between relative tree size and drought-induced mortality is rarely analyzed, maybe considering tree height as a measure of size. Authors could refer to the most recent textbook about Forest Growth Modelling to verify this statement. Plenty of examples of tree mortality dependencies on tree size, for instance, Weiskittel et.al., 2011 - Forest Growth and Yield Modeling or Burkhardt & Tome 2012 - Modeling forest trees and stands.

Response: Thank you for providing the valuable reference. We have rephrased the entire paragraph in response to your comments as well as the comments from the first reviewer.

6. L90. Pre-drought is 2012? Since this is the first time you declare pre-drought you should specify the year.

Response: Yes. We have further clarified that pre-drought is defined as 2012 in Line 105.

7. L113. In Fig2a the units of P(m) don't match to the text (mm) for example line 405. Also, the relationship between T and P doesn't show the drought years claimed, the authors should find a better way to illustrate the diagram.

Response: We have standardized the unit of mean annual total precipitation (MAP) to millimeter (mm) in the main text. Furthermore, to emphasize the mean annual temperature (MAT) and MAP during the drought period and underscore their contrasts with the average conditions of the study site, we have made enhancements to Fig. 2a. This has been accomplished by utilizing solid blue or red lines to represent the MAT and MAP during the drought, dashed blue or red lines to signify the average MAT and MAP of the study site, and employing shaded areas with blue or red

colorations to illustrate the disparities in MAT or MAP between the drought period and the average conditions. The revised Fig. 2 is shown below.

Fig. R9 Tree mortality rate varied by spatial location, genus, and elevation. a, The location of our study area in the southern Sierra Nevada mountains, California, USA and its annual mean temperature (T) and annual total precipitation (P) during the 2012-2016 drought, compared to the average annual mean temperature (T_{ave}) and the average annual total precipitation (P_{ave}) from 1980 to 2019. The shaded areas indicate the differences in precipitation or temperature between the drought period (2012-2016 in solid lines) and averaged conditions (dashed lines). **b,** A map depicting tree mortality rate from 2012 to 2016 across the study area. A tree was identified as dead if over 35% of its crown was classified as dead, following the suggestion from Stovall et al.¹⁷. Tree mortality rate was presented as the percentage of dead trees per $100\text{ m} \times 100\text{ m}$ cell. Polygons with grey boundary lines represent forest stand³⁶. The mean stand size was 0.144 km^2 with a coefficient of variation of 86.6%. **c,** Changes of tree mortality rate by elevation for the four major genera in the study area. The marker size is proportional to the number of trees in each elevation gradient (100 m) for each genus.

8. L122-123. I don't consider that a "consistent positive or consistent negative correlation" with tree size should be expected. Typically, tree mortality has a "U"

relationship with tree size, i.e., tree diameter, in empirical mortality models, using logistic regression. Authors could refer to some studies such as Fortin, et.al., 2008 (doi:10.1051/forest:2007088) and Salas & Weiskittel, 2020 (<https://doi.org/10.1016/j.foreco.2020.118369>).

Response: We highly appreciate your suggestion and the references you provided. To prevent any potential confusion, we have deleted this sentence in the revised manuscript.

9. L131. Delete “for water during drought”.

Response: Deleted as suggested.

10. L176. I don't think that authors have similar findings between competition metrics here. For some species, the results seem quite different depending on the competition index, i.e., CCHH vs CVTH.

Response: We agree with the reviewer's observation that the relationships of CVTH with tree mortality rate diverged from that of CCTH. As pointed out by the reviewer, our analysis uncovered a nonlinear relationship between CVTH and tree mortality for *Abies* trees, while a slight positive association was evident for *Quercus* trees (Supplementary Fig. 12). These distinctions might be attributed to the disparities in their respective definitions. In comparison to CCTH and CC66, which primarily focus on the central tree, CVTH encompasses height variations of all trees within a forest stand (Supplementary Fig. 8). Forest stands with high CVTH may not necessarily exhibit high CCTH and CC66. The unique demographic characteristics of *Abies* and *Quercus* trees may potentially increase the likelihood of observing these mismatches. Specifically, a large proportion of *Abies* trees were situated at seasonally cold-limited high altitudes (> 2000 m), characterized by abundant bedrock outcrops with poor soil conditions, while the majority of *Quercus* trees grew in water-limited low altitudes with hot and dry climates (Supplementary Fig. 1). *Abies* and *Quercus* trees may adapt better to these harsh habitat environments (Dolanc et al. 2013; Stephenson and Das 2020), potentially making them taller than neighboring trees of other genera. As CVTH increases, the chance for *Abies* and *Quercus* trees to be taller than neighboring trees increases. The absence of shadows from taller neighboring trees may therefore lead to an increase in the mortality rate for *Abies* trees at high altitudes and for *Quercus* trees.

We have further clarified the disparities between the relationships of CVTH with tree mortality and that of CCTH in Lines 257-265, and we have discussed the potential reasons leading to these disparities in Lines 265-275. Moreover, we agree with the reviewer that comparing the impacts of different competition indices on tree mortality rate can provide further insights on the results. This point has also been emphasized in Lines 275-278.

11. L406. What do you mean exactly by “abnormally high temperature”? Please be specific and quantify.

Response: Thank you for pointing it out. We have rephrased the sentence to: “The extreme drought was accompanied by significantly high temperatures (0.70 °C higher than the mean annual temperature for 1980-2019; Fig. 2a)⁵⁶”

12. L411. According to figure S6, looks like that a systematic sampling over a regular grid was used. It is not clear how the field plots were established.

Response: The field plot measurements conducted in this study followed a carefully designed stratified sampling approach (Jakubowski et al. 2013). Initially, a random plot location was chosen within the study site. Subsequently, a systematic grid with a spacing of 500 m was overlaid across the study area, with each grid intersection serving as a potential field plot location. If any selected plot encompassed landing, road, river, or otherwise physically inaccessible areas within its 12.62 m radius, it was relocated randomly by 25 m along one of the four cardinal directions. Ultimately, a total of 120 plots were identified using this method, and they were located through a GPS in the field. This stratified sampling strategy aligns with established practices in forest inventory (Jakubowski et al. 2013), and we have provided additional clarity on this process in the Methods section, specifically in Lines 380-388.

13. L413. The authors need to include more details about the field plot measurements. Besides species ID and position, other measures were taken. Did the field crew identify live or dead trees for example?

Response: In addition to the ID, tree species, and tree location, a comprehensive set of attributes was recorded during fieldwork. These attributes included measurements such as tree height, diameter at breast height, height to the base of the live crown, canopy cover, and tree vigor. However, except for tree height, other attributes were not used in the current study. Therefore, we refrained from providing extensive information to enhance the clarity of the study. We have added more details about the

tree height measurement process in Lines 396-402. For readers seeking information on these field measurements, we have included a reference (Jakubowski et al. 2013) in the revised manuscript.

Furthermore, it is worth mentioning that the identification of live or deceased trees was facilitated through tree vigor records, as discussed earlier. However, these field measurements were conducted in 2008, a period preceding the occurrence of the drought under investigation. Consequently, these records were not integrated into the present study's analysis. As for the accurate identification of ground-truth dead trees, this was accomplished through meticulous visual assessment of high-resolution aerial imagery, as elaborated in Lines 466-472 and Supplementary Box 3.

14. L414 and 445. Not minor details are missing here. What is the precision of the tree location GPS data? For how long time the GPS placed to get the location of each tree? Did the tree location procedure include a GPS post-processed correction? Those are important details that should be included, to have more information about the precision in the co-registration data of field plots, lidar, and aerial imagery. I consider that the authors should extend the details as they did with the tree segmentation, species classification, and dead tree identification in boxes 1, 2, and 3.

Response: Thank you for the valuable comments. In this study, we first utilized a Trimble GeoXH Global Positioning System (GPS) receiver to measure plot centers' locations. To minimize any potential multipath issues, a Trimble Zephyr antenna was positioned at a height of 3 m, employing an antenna pole. The implementation of this setup was complemented by the availability of Continuously Operating Reference Stations and University NAVSTAR Consortium stations for differential GPS post-processing, all located within a radius of fewer than 20 km from the designated plots. Throughout the data collection phase, stringent efforts were devoted to ensuring a low positional dilution of precision (PDOP) value, rigorously maintained below 5. In instances where the PDOP value exceeded this threshold, immediate measures were taken to relocate the GPS receiver to a more open forest canopy location, typically up to 30 meters away. To bolster the accuracy of each GPS position determination, a minimum of 300 measurements were captured at one-second intervals for every position. It is noteworthy that the majority of positions encompassed over 1,000 measurements, with some instances recording as many as 7,700 measurements. The culmination of these measures yielded a centimeter-level precision accuracy in the deamination of plot centers' locations. Once the precise coordinates of plot centers

were established, we then employed an Impulse laser ranger finder and an Impulse electronic compass to measure the distance and angle from the plot center to each individual tree, thereby georeferencing all trees. We have further clarified the protocol of field measurements in the Methods section in Lines 387-402.

15. L419-420. Can authors be more specific on the “standardized procedure” used? Criteria to define and remove outliers, filtering and attributes for normalization.

Response: We have provided details for each step of this standardized lidar data processing procedure in Lines 410-421 as follows.

“Outlier removal aims to mitigate the influence of noise points arising from wind, high-flying objects, and the multi-path effect. To achieve this, a distance-based method was employed, identifying noise points by assessing whether the average distance between a point and its ten closest neighboring points exceeds a threshold of $\mu+5\sigma$ (where μ and σ represent the mean and standard deviation of point distances, respectively)⁶⁰. Filtering, the subsequent step, serves the purpose of separating ground points from non-ground points, facilitating the generation of terrain elevation products. In this study, we employed an enhanced progressive triangulated irregular network densification filtering algorithm⁶¹. Normalization, the final step, assumes the role of counteracting the influence of terrain elevation on lidar height measurements, which accomplished by subtracting the elevation of a point by its corresponding ground elevation. All the aforementioned preprocessing steps were executed within the LiDAR360 software (GreenValley International Inc.), employing the default parameter settings.”

16. L424. How many points did the authors use? Please add the location of the points in Figure S6.

Response: A total of 69 tie points were used in this study. We have further clarified this in Lines 425-427 and added their locations in Supplementary Fig. 13.

17. L428-429. I suppose the authors used the same lidar and aerial imagery here, please clarify.

Response: Yes, you are correct. The same lidar and aerial imagery data were used here. We have revised the texts in Lines 430-431 to clarify this.

18. L431. How TWI and solar energy were derived from DEM?

Response: Both TWI and solar radiation are variables influenced by terrain elevations.

TWI is calculated as a function of both the slope and the upstream contributing area per unit width orthogonal to the flow direction (Sørensen et al. 2006), and solar radiation is calculated as a function of latitude, date, within-day time, and terrain elevation, aspect, and slope (Ruiz-Arias et al. 2009). In this study, TWI was calculated from the lidar-derived digital terrain model (DTM) using the TWI tool in SGAG-GIS, and the hourly solar radiation from 10 am to 2 pm on August 1st, 2016 were calculated from the lidar-derived DTM using the Area Solar Radiation tool in ArcGIS. We have further clarified this in Lines 431-440.

19. L472. How do authors define the 15 m radius? Seems that is an arbitrary value that is properly justified. Looking at the distribution of tree height by species in Fig S1, there are some clear differences in the mean and distribution in tree height among species. The definition of the neighboring trees (competitive trees) certainly modifies the values of the CCTH, CC66, and CVTH used to evaluate the competition. Please, include an informative justification of the radius of provide evidence about the sensitivity of the competition metrics to the radius definition.

Response: Thank you for the valuable suggestion. In response, we conducted a sensitivity analysis on the relationships between tree mortality and CCTH concerning the size of the neighborhood used for calculating CCTH. To achieve this, we calculated CCTH using four different neighborhood sizes with radii of 15 m, 30 m, 50 m, and 100 m, respectively, and evaluated their relationships with tree mortality. Our results showed that while CCTH variations reduced as the neighborhood size increased, the calculated CCTH of different neighborhood sizes exhibited consistently negative relationships with tree mortality (Fig. R10), demonstrating the robustness of the findings of this study. Given the fact that we have found a radius of 15 m can effectively capture the competition among trees in our previous study (Ma et al. 2017), we decided to continue to use 15 m in the current study. Nevertheless, we recognize the importance of justifying the selection of the neighborhood size, and have added the corresponding discussion to the manuscript in Lines 251-253, Lines 496-499, Supplementary Fig. 7, and Supplementary Table 11.

Fig. R10 Sensitivity analysis of the relationships between CCTH and tree mortality rate during the 2012-2016 drought, considering the influence of the neighborhood size used to calculate CCTH. CCTH was calculated using four different neighborhood sizes, with radii of 15 m, 30 m, 50 m, and 100 m, respectively. Forest stands were used as the basic statistical units here. Tree mortality rate within each forest stand was binned by CCTH with an interval of 10%, and the linear regression method weighted by the number of trees in each bin was used to fit their relationships. Red dots in all panels represent the average tree mortality within each tree height or CCTH bin. Only half of the bins were presented for visual clarity. The size of red dots is proportional to the number of trees in each bin. Solid lines are fitted lines with a $P < 0.05$. The corresponding statistics of each regression segments are presented in Supplementary Table 11.

20. 505. evapotranspiration (ET).

Response: The abbreviation ET has been defined in Line 51.

21. Box 1. “Based on field measurements, segmented trees with a too small or a too large ($>200 \text{ m}^2$) crown area were probably caused by noise or mis-segmentation”.

How much is too small, it is weird that there is a reference value for too large, but is missing for too small.

Response: In this study, we defined a tree crown with an area less than 1 m^2 as being categorized as “too small”. We have clarified this as follows in Supplementary Box 1.

“Based on field measurements, tree segments with crown areas that were either too small ($< 1 \text{ m}^2$) or too large ($> 200 \text{ m}^2$) were likely attributed to noise or mis-segmentation.”

22. Box 2. “Field records of individual trees were matched with lidar-derived individual tree crowns based on their spatial locations, and tree height and positioning

uncertainty were also considered as constraints following the procedure used in Ma et al. (1)". Please summarise how you consider the position uncertainty, given the interval between field ground survey and lidar acquisition.

Response: We have summarized the steps involved in matching tree locations in Supplementary Box 2, as presented below.

"These field-recorded trees were matched with lidar-derived individual tree crowns based on their spatial locations, with tree height also considered a constraint, following the approach outlined in Ma et al. (1). Specifically, we began by generating a 2-m buffer around the location of each field-recorded tree and retained all lidar-segmented trees within the buffer as potential match candidates. Then, differences between field and lidar tree height measurements were assessed. If the height difference fell within the range of -1 m to 5 m, the corresponding lidar-derived tree segment was retained as a potential match candidate; otherwise, it was excluded. In cases where multiple potential match candidates persisted even after this refinement, the lidar-derived tree segment with the least disparity in tree height was selected as the final match. These tree matching steps accounted for potential positioning errors in field measurements and discrepancies between field-recorded and lidar-derived tree positions (tree base vs. treetop)."

References

- Dolanc, C.R., Thorne, J.H., & Safford, H.D. (2013). Widespread shifts in the demographic structure of subalpine forests in the Sierra Nevada, California, 1934 to 2007. *Global Ecology and Biogeography*, 22, 264-276
- Jakubowski, M.K., Guo, Q., Collins, B., Stephens, S., & Kelly, M. (2013). Predicting surface fuel models and fuel metrics using Lidar and CIR imagery in a dense, mountainous forest. *Photogrammetric Engineering & Remote Sensing*, 79, 37-49
- Ma, Q., Su, Y., Tao, S., & Guo, Q. (2017). Quantifying individual tree growth and tree competition using bi-temporal airborne laser scanning data: a case study in the Sierra Nevada Mountains, California. *International Journal of Digital Earth*, 11, 485-503
- Ruiz-Arias, J.A., Tovar-Pescador, J., Pozo-Vázquez, D., & Alsamamra, H. (2009). A comparative analysis of DEM-based models to estimate the solar radiation in mountainous terrain. *International Journal of Geographical Information Science*, 23, 1049-1076
- Sörensen, R., Zinko, U. & Seibert, J. (2006) On the calculation of the topographic

wetness index: evaluation of different methods based on field observations.
Hydrology and Earth System Sciences, 10, 101-112.

Stephenson, N.L., & Das, A.J. (2020). Height-related changes in forest composition explain increasing tree mortality with height during an extreme drought. *Nature Communications*, 11, 3402

Stovall, A.E.L., Shugart, H., & Yang, X. (2019). Tree height explains mortality risk during an intense drought. *Nature Communications*, 10, 4385

REVIEWER COMMENTS

Reviewer #1 (Remarks to the Author):

Well done. The authors did a fantastic job addressing reviewer comments. No further suggestions or comments from me.

I recommend this manuscript for publication.

Reviewer #2 (Remarks to the Author):

This marks my second evaluation of this insightful manuscript, and I must acknowledge the authors' efforts in implementing my recommendations, along with those from the other reviewer. I appreciate their hard work in rephrasing the content, resulting in significantly improved clarity and enhanced readability. The inclusion of the additional information requested helps me to better understand the methods and the analytical approach. In my opinion, the manuscript now constitutes a significant piece of work deserving publication.

However, I still have concerns regarding the mixed linear model analyses that might need some attention before publishing this work. The authors "conducted mixed linear model analyses to assess the correlations between neighborhood canopy structural attributes while factoring in the influence of tree height." To do so, in each model, neighborhood canopy structural traits (represented by CCTH) were treated as the fixed effect, while tree height was incorporated as the random effect in three stratified groups of tree height (i.e., low, medium, high).

Despite the authors acknowledging the fact that "additional analyses to explore the relationships between tree mortality and neighborhood canopy structural attributes, while accounting for the potential impact of individual tree size, would bolster the strength of our results" in their response letter, the linear mixed model used and the results presented didn't support their inference. First, table S12 only shows the slope and P-value of the fixed effect coefficient, completely omitting the variance components of the random effects. Moreover, the results in table S12 don't support the claim that "relationships between CCTH and tree mortality rate were independent of tree height" (L254-256).

Methodologically, including the height levels (as groups of 33.3% for each class in height gradient) makes no sense for controlling the effect of tree size and then evaluating the effect of neighborhood canopy structural attributes. Instead, including tree height as a random effect will allow relaxing the assumption of independence of residuals, as they would absorb the unaccounted variability at the aforementioned levels, which is not clear if the random effects were included in the intercept or slope of the model.

I think the authors are misinterpreting the results from the mixed effect models. To conclude that "the mixed linear modeling analyses did not yield alterations in the observed correlations," the significance of the random effect should be tested first based on the likelihood ratio test ($p < 0.05$) between nested models (only fixed vs. fixed + randoms).

Response to Reviewers

Reviewer #1

Well done. The authors did a fantastic job addressing reviewer comments. No further suggestions or comments from me. I recommend this manuscript for publication.

Response: Thank you for your positive feedback. We sincerely appreciate your previous invaluable comments, which were very helpful for us to improve the quality of the manuscript.

Reviewer #2

1. This marks my second evaluation of this insightful manuscript, and I must acknowledge the authors' efforts in implementing my recommendations, along with those from the other reviewer. I appreciate their hard work in rephrasing the content, resulting in significantly improved clarity and enhanced readability. The inclusion of the additional information requested helps me to better understand the methods and the analytical approach. In my opinion, the manuscript now constitutes a significant piece of work deserving publication.

Response: Your positive feedback is truly valued. In response to your concern regarding the mixed linear effect modeling analyses, we have further revised the manuscript. Please find detailed responses below, and all the revisions in the manuscript have been highlighted in yellow for your convenience.

2. However, I still have concerns regarding the mixed linear model analyses that might need some attention before publishing this work. The authors “conducted mixed linear model analyses to assess the correlations between neighborhood canopy structural attributes while factoring in the influence of tree height.” To do so, in each model, neighborhood canopy structural traits (represented by CCTH) were treated as the fixed effect, while tree height was incorporated as the random effect in three stratified groups of tree height (i.e., low, medium, high).

Despite the authors acknowledging the fact that “additional analyses to explore the relationships between tree mortality and neighborhood canopy structural attributes, while accounting for the potential impact of individual tree size, would bolster the strength of our results” in their response letter, the linear mixed model used and the

results presented didn't support their inference. First, table S12 only shows the slope and P-value of the fixed effect coefficient, completely omitting the variance components of the random effects. Moreover, the results in table S12 don't support the claim that "relationships between CCTH and tree mortality rate were independent of tree height" (L254-256).

Methodologically, including the height levels (as groups of 33.3% for each class in height gradient) makes no sense for controlling the effect of tree size and then evaluating the effect of neighborhood canopy structural attributes. Instead, including tree height as a random effect will allow relaxing the assumption of independence of residuals, as they would absorb the unaccounted variability at the aforementioned levels, which is not clear if the random effects were included in the intercept or slope of the model.

I think the authors are misinterpreting the results from the mixed effect models. To conclude that "the mixed linear modeling analyses did not yield alterations in the observed correlations," the significance of the random effect should be tested first based on the likelihood ratio test ($p < 0.05$) between nested models (only fixed vs. fixed + random).

Response: We agree with the reviewer's observation that the existing mixed linear modeling analysis results do not support the claim that "relationships between CCTH and tree mortality rate were independent of tree height". We apologize for any confusion caused by the inaccurate phrasing. Our intention was to assess the consistency of the negative correlation between CCTH and tree mortality across various tree height groups. Our linear mixed models did prove this point across all tree genera. To prevent misinterpretation of the mixed linear modeling analysis results, we have revised the sentence in Line 255-257.

Additionally, in response to the reviewer's feedback, we have enhanced the methodology by refining the mixed linear effect models. Instead of employing grouped tree height levels, we incorporated continuous tree height measurements as the random effect. Furthermore, to assess the significance of the random effect, we compared the mixed linear effect model for all genera or each genus with its corresponding model that did not account for tree height as the random effect, using analysis of variance. The results, presented in Table R1, demonstrated significant differences between models with and without the random effect, elucidating the

significance of the random effect in all mixed linear effect models. These findings affirm that the mixed linear modeling analyses did not yield alterations in the observed correlations between CCTH and tree mortality. We have provided updated results of the mixed linear effect modeling analyses in Line 521-528 and Supplementary Table 12.

Table R1. Statistics of relationships between CCTH and tree mortality rate during the 2012-2016 drought, considering the confounding influence of tree height. Tree mortality rate within each forest stand was binned by CCTH and tree height with an interval of 4% and 5 m. The slope and significance (represented by *P*) of the fixed effect (CCTH) and the significance (represented by *P*) of the random effect (tree height) were reported. The significance of the random effect was determined by comparing the mixed linear effect model for all genera or each genus with its corresponding model that did not account for tree height as the random effect, using analysis of variance.

Genus	Fixed effect: CCTH		Random effect: Tree height
	Slope	P	P
All	-0.007	<0.001	<0.001
Abies	-0.011	<0.001	<0.001
Cedrus	-0.005	<0.001	<0.001
Pinus	-0.009	<0.001	<0.001
Quercus	-0.014	<0.001	<0.001